# Multi-Fidelity Multi-Armed Bandits Revisited

**Xuchuang Wang**
Chinese University of Hong Kong
xcwang@cse.cuhk.edu.hk

**Qingyun Wu**
Pennsylvania State University
qingyun.wu@psu.edu

**Wei Chen**
Microsoft Research
weic@microsoft.com

**John C.S. Lui**
Chinese University of Hong Kong
cslui@cse.cuhk.edu.hk

## Abstract

We study the multi-fidelity multi-armed bandit (`MF-MAB`), an extension of the canonical multi-armed bandit (MAB) problem. `MF-MAB` allows each arm to be pulled with different costs (fidelities) and observation accuracy. We study both the best arm identification with fixed confidence (`BAI`) and the regret minimization objectives. For `BAI`, we present (a) a cost complexity lower bound, (b) an algorithmic framework with two alternative fidelity selection procedures, and (c) both procedures' cost complexity upper bounds. From both cost complexity bounds of `MF-MAB`, one can recover the standard sample complexity bounds of the classic (single-fidelity) MAB. For regret minimization of `MF-MAB`, we propose a new regret definition, prove its problem-independent regret lower bound $\Omega(K^{1/3}\Lambda^{2/3})$ and problem-dependent lower bound $\Omega(K \log \Lambda)$, where $K$ is the number of arms and $\Lambda$ is the decision budget in terms of cost, and devise an elimination-based algorithm whose worst-cost regret upper bound matches its corresponding lower bound up to some logarithmic terms and, whose problem-dependent bound matches its corresponding lower bound in terms of $\Lambda$.

## 1 Introduction

The multi-armed bandits (MAB) problem was first introduced in the seminal work by Lai and Robbins [22] and was extensively studied (ref. Bubeck and Cesa-Bianchi [4], Lattimore and Szepesvári [24]). In stochastic MAB, the decision maker repeatedly pulls arms among a set of $K \in \mathbb{N}^+$ arms and observes rewards drawn from unknown distributions of the pulled arms. The initial objective of MAB is *regret minimization* [22, 1], where the regret is the cumulative differences between the rewards from pulling the optimal arm and that from the concerned algorithm's arm pulling strategy. This is a fundamental sequential decision making framework for studying the exploration-exploitation trade-off, where one needs to balance between optimistically exploring arms with high uncertainty in reward (exploration) and myopically exploiting arms with high empirical reward average (exploitation). Another task of MAB is the best arm identification (`BAI`, a.k.a., pure exploration), later introduced by Even-Dar et al. [7, 8], Mannor and Tsitsiklis [27]. Best arm identification aims to find the best arm *without* considering the cumulative regret (reward) during the learning process. In this paper, we focus on the fixed confidence case, with the goal of identifying the best arm with a fixed confidence with as few number of decision rounds as possible.

In this paper, we investigate a generalized MAB model with a wide range of real-world applications—the *multi-fidelity multi-armed bandit* (`MF-MAB`)—introduced by Kandasamy et al. [19]. We study both the regret minimization and best arm identification objectives under the `MF-MAB` model. The `MF-MAB` model introduces flexibility for exploring arms: instead of squarely pulling an arm to observe its

37th Conference on Neural Information Processing Systems (NeurIPS 2023).

reward sample as in MAB, MF-MAB offers $M \in \mathbb{N}^+$ different accesses to explore an arm, called $M$ fidelities. Pulling an arm $k \in \mathcal{K} := \{1, \dots, K\}$ at fidelity $m \in \mathcal{M} := \{1, \dots, M\}$, the decision maker pays a cost of $\lambda^{(m)}$ and observes a reward $X_k^{(m)}$ whose mean $\mu_k^{(m)}$ is not too far away from the arm's true reward mean $\mu_k$. More formally, $|\mu_k^{(m)} - \mu_k| \leqslant \zeta^{(m)}$ where $\zeta^{(m)} \geqslant 0$ is the error upper bound at fidelity $m$. The formal definition of MF-MAB is presented in Section 2.

## 1.1 Contributions

In Section 3, we first look into the best arm identification with fixed confidence (BAI) objective in MF-MAB, which has wide applications in hyperparameter optimization [15, §1.4] and neural architecture search (NAS) [6]. In the context of NAS, each arm of MF-MAB corresponds to one configuration of neural network architecture, and the different fidelities of pulling this arm correspond to training the neural network of this configuration with different sample sizes of training data (see details in Section 2.1). As in MF-MAB pulling an arm at different fidelities has different costs, we consider the total cost needed by an algorithm and refer to it as *cost complexity* (a generalization of sample complexity). We first derive a cost complexity lower bound $\Omega(\sum_{k \in \mathcal{K}} \min_{m:\Delta_k^{(m)}>0} (\lambda^{(m)}/(\Delta_k^{(m)})^2) \log \delta^{-1})$, where the $\Delta_k^{(m)}$ is the reward gap of arm $k$ at fidelity $m$ (defined in Eq.(1)) and $\delta$ is the confidence parameter. We then devise a Lower-Upper Confidence Bound (LUCB) algorithmic framework with two different fidelity selection procedures and prove both procedures' cost complexity upper bounds. The first procedure focuses on finding the optimal fidelity suggested by the minimization in lower bound via an upper confidence bound (UCB) algorithm, which may pay high additional costs during searching the optimal fidelities. The second procedure, instead of identifying the optimal fidelity, seeks for good fidelities that are at least half as good as the optimal ones so as to reduce the cost for identifying the optimal fidelity while still enjoying fair theoretical performance.

In Section 4, we next study the regret minimization objective in MF-MAB. We introduce a novel definition of regret, wherein the rewards obtained are dependent on the pulled arms, while the selected fidelities solely affect the accuracy of the observed rewards. The new regret definition covers applications like product management and advertisement distribution [13, 10] where the distributed advertisement (ad) determines the actual (but unknown) reward while the cost (fidelity) spent by the company on evaluating the impact of this ad decides the accuracy of the observed rewards (see Remark 4.1). The difference of our regret definition and the one studied by Kandasamy et al. [19] is discussed in Remark 4.2. We first propose both a problem-independent (a.k.a., worst-case) regret lower bound $\Omega(K^{1/3}\Lambda^{2/3})$ and a problem-dependent regret lower bound $\Omega(K \log \Lambda)$ where the $\Lambda \, (> 0)$ is the decision budget. We then devise an elimination-based algorithm, which explores (and eliminates) arms in the highest fidelity $M$ and exploits the remaining arms in the lowest fidelity 1. The algorithm enjoys a $\tilde{O}(K^{1/3}\Lambda^{2/3})$ problem-independent regret upper bound, which matches the corresponding lower bound up to some logarithmic factor, and also a problem-dependent bound matching its corresponding lower bound tightly in a class of MF-MAB instances.

## 1.2 Related Works

Multi-fidelity multi-armed bandits (MF-MAB) was first proposed by Kandasamy et al. [19]. They studied a cumulative regret minimization setting whose regret definition is different from ours (see Remark 4.2 for more details). Later, Kandasamy et al. [18] extended MF-MAB to bandits optimization, i.e., in continuous action space, with the objective of minimizing simple regret, and Kandasamy et al. [20] further extended Kandasamy et al. [18] to the continuous fidelity space with the same objective of minimizing simple regret. We note that the simple regret minimization in bandits optimization corresponds to the best arm identification with fixed budget in multi-armed bandits, and, therefore, is different from the cumulative regret minimization and best arm identification with fixed confidence objectives studied in this paper. During the submission of this work, the authors were aware that the BAI task under MF-MAB model was also studied by Poiani et al. [30], where they proposed a similar cost complexity lower bound to our Theorem 3.1 and their proposed algorithm is similar to our EXPLORE-C procedure in Appendix D. Except for these two results, all other results proposed in this paper, including two BAI algorithms in Section 3.2 and the regret minimization results in Section 4, are novel.

The multi-fidelity optimization was first introduced in simulating expensive environments via cheap surrogate models [14, 11], and, later on, was extended to many real-world applications, e.g., shape design optimization for ship [3] or wing [32], and wind farm optimization [31]. Recently, the multi-fidelity idea was employed in hyperparameter optimization (HPO) for automated machine learning (autoML) [16, 26, 9, 25]. The fidelity may correspond to training time, data set subsampling, and feature subsampling, etc. These multi-fidelity settings help agents to discard some hyperparameter configurations with low cost (a.k.a., early discarding [28, §3.1.3]). Jamieson and Talwalkar [16] modeled hyperparameter optimization as non-stochastic best arm identification and applied the successive halving algorithm (SHA) to address it. Li et al. [26] introduced Hyperband, a hyperparameter configuring and resources allocation method which employed SHA as its subroutine to solve real HPO tasks. Falkner et al. [9] proposed BOHB, which use Bayesian optimization method to select and update hyperparameters and employ SHA to allocate resources. Li et al. [25] extended SHA to asynchronous case for parallel hyperparameter optimization. This line of works was based on the non-stochastic best arm identification problem [16] and, therefore, is very different from our stochastic modelling.

## 2 Model

We consider a $K$ ($\in \mathbb{N}^+$)-armed bandit. Each arm can be pulled with $M$ ($\in \mathbb{N}^+$) different fidelities. When an arm $k \in \mathcal{K} \coloneqq \{1, \ldots, K\}$ is pulled at fidelity $m \in \mathcal{M} \coloneqq \{1, \ldots, M\}$, one pays a cost $\lambda^{(m)}$ ($> 0$) and observes a feedback value drawn from a $[0, 1]$-supported probability distribution with mean $\mu_k^{(m)}$. We assume the non-trivial case that $|\mu_k^{(m)} - \mu_k^{(M)}| \leqslant \zeta^{(m)}$, where $\zeta^{(m)} \geqslant 0$ is the observation error upper bound at fidelity $m$. Without loss of generality, we assume $\lambda^{(1)} \leqslant \ldots \leqslant \lambda^{(M)}$; otherwise, we can relabel the fidelity so that the $\lambda^{(m)}$ is non-decreasing with respect to $m$. We call the reward mean of an arm at the highest fidelity $M$ as this arm's *true reward mean* and, without loss of generality, assume that these true reward means are in a descending order $\mu_1^{(M)} > \mu_2^{(M)} \geqslant \mu_3^{(M)} \geqslant \ldots \geqslant \mu_K^{(M)}$, and arm 1 is the unique optimal arm.

For the online learning problem, we assume that the reward distribution and the mean $\mu_k^{(m)}$ of each arm $k$ at fidelity $m$ are unknown, while the costs $\lambda^{(m)}$'s and the error upper bounds $\zeta^{(m)}$'s are known to the learning agent.[1] To summarize, a MF-MAB instance $\mathcal{I}$ is parameterized by a set of tuples $(\mathcal{K}, \mathcal{M}, (\lambda^{(m)})_{m \in \mathcal{M}}, (\zeta^{(m)})_{m \in \mathcal{M}}, (\mu_k^{(m)})_{(k,m) \in \mathcal{K} \times \mathcal{M}})$, with elements described above.

In this paper, we consider two tasks on the above multi-fidelity bandit model: best arm identification and regret minimization. We will define these two tasks in the respective sections below.

### 2.1 Application

One typical application of the best arm identification problem is hyperparameter optimization [15, 6] (including neural architecture search) for machine learning, in which the goal is to identify the best hyperparameter configuration—the training set-up for a machine learning model attaining the best predictive performance—with as low resource as possible. A mapping between concepts in this application and MF-MAB is discussed as follows. **Arm:** Hyperparameter configurations of machine learning models, e.g., neural network architectures. **Reward:** Predictive performance of the resulting machine learning model trained based on the selected configuration (arm). **Fidelity dimension:** For a particular hyperparameter configuration (arm), one typically has the choice to determine a certain level of resources to allocate for training the model. The concept of "training resource" can be considered the fidelity dimension. More concretely, commonly used training resources include *the number of epochs* and *the training data sample*, both of which satisfy our cost assumption. For example, the larger the number of epochs or training data samples, the more expensive to train the model.

**Remark 2.1** (On the assumptions of multi-fidelity feedback in the hyperparameter optimization application). Observation error upper bound $\zeta^{(m)}$ is the maximum distance from resources allocated to the terminal validation loss. According to benchmarked results in two recent benchmarks for multi-fidelity hyperparameter optimization, including HPOBench [5] and YAHPO Gym [29], under the typically used fidelity dimension, including the number of epochs, the training data sample, etc., the

---

[1]We provide application examples where $\lambda^{(m)}$ and $\zeta^{(m)}$ are known in Section 2.1

maximum distance from the terminal validation loss often decreases with the increase of resources, i.e., $\zeta^{(m)}$ decreases with the increase of $m$. Thanks to these benchmarks, it is also convenient to know $\zeta^{(m)}$ under different fidelities $m$ for commonly used types of fidelity dimension.

# 3 Best Arm Identification with Fixed Confidence

In this section, we study the best arm identification with fixed confidence (BAI) task in the multi-fidelity multi-armed bandits (MF-MAB) model. The objective of BAI is to minimize the total budget spent for identifying the best arm with a confidence of at least $1 - \delta$. As the cost of pulling an arm in MF-MAB depends on the chosen fidelity, we use the total cost, instead of total pulling times (sample complexity), as our criterion, and refer to it as *cost complexity*. We first present a cost complexity lower bound in Section 3.1, then propose a LUCB algorithmic framework with two alternative procedures in Section 3.2, and analyze their cost complexity upper bounds in Section 3.3. Lastly, we introduce concrete applications of the BAI problem in Section 2.1.

## 3.1 Cost Complexity Lower Bound

We present the cost complexity lower bound in Theorem 3.1. Its proof is deferred to Appendix E.1. During the submission of this paper, we notice that a similar cost complexity lower bound was also proposed by Poiani et al. [30, Theorem 1].

**Theorem 3.1** (Cost complexity lower bound). *For any algorithm addressing the best arm identification with fixed confidence $1 - \delta$ for any parameter $\delta > 0$, any number of arms $K$, any number of fidelities $M$ with any observation error upper bound sequence $(\zeta^{(1)}, \zeta^{(2)}, \ldots, \zeta^{(M)})$ $(\zeta^{(M)} = 0)$ and any cost sequence $(\lambda^{(1)}, \ldots, \lambda^{(M)})$, and any $K$ fidelity subsets $\{\mathcal{M}_1, \mathcal{M}_2, \ldots, \mathcal{M}_K\}$ where the $\mathcal{M}_k$ is a subset of full fidelity set $\mathcal{M}$ containing the highest fidelity $M$, i.e., $M \in \mathcal{M}_k \in 2^{\mathcal{M}}$, for all $k \in \mathcal{K}$, there exists a MF-MAB instance such that*

$$\mathbb{E}[\Lambda] \geqslant \left( \min_{m \in \mathcal{M}_1} \frac{\lambda^{(m)}}{\mathrm{KL}(\nu_1^{(m)}, \nu_2^{(M)} + \zeta^{(m)})} + \sum_{k \neq 1} \min_{m \in \mathcal{M}_k} \frac{\lambda^{(m)}}{\mathrm{KL}(\nu_k^{(m)}, \nu_1^{(M)} - \zeta^{(m)})} \right) \log \frac{1}{2.4\delta},$$

*where $\mathrm{KL}$ is the KL-divergence between two probability distributions, $\nu_k^{(m)}$ is the probability distribution associated with arm $k$ when pulled at fidelity $m$, and $\nu \pm \zeta$ means to positively/negatively shift the distribution $\nu$ by an offset $\zeta$.*

According to the KL-divergences' two inputs in the above lower bound, we define the reward gap $\Delta_k^{(m)}$ of arm $k$ at fidelity $m$ as follows,

$$\Delta_k^{(m)} := \begin{cases} \mu_1^{(M)} - (\mu_k^{(m)} + \zeta^{(m)}) & \forall k \neq 1 \\ (\mu_1^{(m)} - \zeta^{(m)}) - \mu_2^{(M)} & \forall k = 1 \end{cases}. \tag{1}$$

For any suboptimal arm $k \neq 1$, the gap $\Delta_k^{(m)}$ quantifies the distance between the optimal arm's reward mean $\mu_1^{(M)}$ and the suboptimal arm's reward mean upper bound at fidelity $m$, i.e., $\mu_k^{(m)} + \zeta^{(m)}$; and for the optimal arm 1, the gap $\Delta_1^{(m)}$ represents the distance between the optimal arm's reward mean lower bound at fidelity $m$, i.e., $\mu_1^{(m)} - \zeta^{(m)}$, and the second-best arm's reward mean $\mu_2^{(M)}$.

**Remark 3.2.** If all reward distributions are assumed to be Bernoulli or Gaussian and let $\mathcal{M}_k = \{m : \Delta_k^{(m)} > 0\}$, the regret bound can be simplified as

$$\mathbb{E}[\Lambda] \geqslant C \sum_{k \in \mathcal{K}} \min_{m : \Delta_k^{(m)} > 0} \frac{\lambda^{(m)}}{(\Delta_k^{(m)})^2} \log \frac{1}{\delta}, \tag{2}$$

where $C > 0$ is a constant depending on the reward distributions assumed. Especially, we denote $m_k^* := \arg \min_{m : \Delta_k^{(m)} > 0} \frac{\lambda^{(m)}}{(\Delta_k^{(m)})^2}$, and with some algebraic transformations, it can be expressed as

$$m_k^* = \arg \max_{m \in \mathcal{M}} \frac{\Delta_k^{(m)}}{\sqrt{\lambda^{(m)}}}. \tag{3}$$

---

**Algorithm 1** LUCB Framework for Multi-Fidelity BAI

---

1: **Input:** violation probability $\delta$
2: **Initialization:** $\hat{\mu}_{k,t}^{(m)} = 0$, $N_{k,t}^{(m)} = 0$, $\text{UCB}_{k,t} = 1$, $\text{LCB}_{k,t} = 0$ for all arm $k$ and fidelity $m$, and
   $\ell_t = 1, u_t = 2, t = 1, \tilde{\mu}_1^{(M)}, \tilde{\mu}_2^{(M)}$.
3: **while** $\text{LCB}_{\ell_t,t} \leqslant \text{UCB}_{u_t,t}$ **do**
4:     $\ell_t \leftarrow \underset{k \in \mathcal{K}}{\arg\max}\, \text{UCB}_{k,t}$, $u_t \leftarrow \underset{k \in \mathcal{K} \setminus \{\ell_t\}}{\arg\max}\, \text{UCB}_{k,t}$     ▷ Select top two UCB indices
5:     EXPLORE($u_t$) and EXPLORE($\ell_t$)     ▷ Any procedure of Algorithm 2
6: **Output:** arm $\ell_t$

---

With $m_k^*$, the lower bound in Eq.(2) can be rewritten as $\mathbb{E}[\Lambda] \geqslant C \sum_{k \in \mathcal{K}} (\lambda^{(m_k^*)}/(\Delta_k^{(m_k^*)})^2) \log(1/\delta)$, and we define the coefficient as $H := \sum_{k \in \mathcal{K}} \lambda^{(m_k^*)}/(\Delta_k^{(m_k^*)})^2$. The $m_k^*$ can be interpreted as the *optimal (most efficient) fidelity* for exploring arm $k$. We note that the current lower bound in Eq.(2) does not contain the cost of finding this $m_k^*$. This cost can be observed in our algorithm's cost complexity upper bound stated in Section 3.3.

## 3.2 Algorithm Design

In this subsection, we propose a Lower-Upper Confidence Bound (LUCB) algorithmic framework with two alternative procedures which employ different mechanisms to select suitable fidelities for arm exploration. Generalized from the original (single-fidelity) LUCB algorithm [17], the LUCB algorithmic framework in §3.2.1 determines two critical arms (the empirical optimal arm $\ell_t$ and second-best arm $u_t$) for exploration in each time slot $t$. However, in MF-MAB, with the critical arms suggested by the LUCB framework, one still needs to decide the fidelities for exploring the critical arms. This fidelity selection faces an accuracy-cost trade-off, i.e., higher fidelity (accuracy) but suffering higher cost, or lower fidelity (accuracy) but enjoying lower cost. This trade-off is different from the common exploration-exploitation trade-off in classic bandits. Because the accuracy and costs are two orthogonal metrics, while in exploration-exploitation trade-off, there is only a single regret metric. We address the accuracy-cost trade-off in §3.2.2 with two alternative procedures.

### 3.2.1 LUCB Algorithmic Framework

The main idea of LUCB [17] is to repeatedly select and explore two critical arms, that is, the empirical optimal arm $\ell_t$ and the empirical second-best arm $u_t$. When both critical arms' confidence intervals are separated—$\ell_t$'s LCB (lower confidence bound) is greater than $u_t$'s UCB (upper confidence bound), the algorithm terminates and outputs the estimated optimal arm $\ell_t$.

The LUCB framework depends on a set of meaningful confidence intervals of the arms' rewards. Usually, the confidence interval of an empirical mean estimate $\hat{\mu}_k^{(m)}$ can be expressed as $(\hat{\mu}_k^{(m)} - \beta(N_{k,t}^{(m)}, t, \delta), \hat{\mu}_k^{(m)} + \beta(N_{k,t}^{(m)}, t, \delta))$, where $\beta(N_{k,t}^{(m)}, t, \delta)$ is the confidence radius and $N_{k,t}^{(m)}$ is the number of times of pulling arm $k$ at fidelity $m$ up to time $t$ (include $t$). As MF-MAB assumes that $|\mu_k^{(m)} - \mu_k^{(M)}| \leqslant \zeta^{(m)}$, based on observations of fidelity $m$, the upper and lower confidence bounds for arm $k$'s true reward mean at the highest fidelity $\mu_k^{(M)}$ can be expressed as follows,

$$\text{UCB}_{k,t}^{(m)} := \hat{\mu}_{k,t}^{(m)} + \zeta^{(m)} + \beta(N_{k,t}^{(m)}, t, \delta), \quad \text{LCB}_{k,t}^{(m)} := \hat{\mu}_{k,t}^{(m)} - \zeta^{(m)} - \beta(N_{k,t}^{(m)}, t, \delta), \quad (4)$$

where we set the confidence radius $\beta(n, t, \delta) = \sqrt{\log(Lt^4/\delta)/n}$ and $L\,(>0)$ is a factor. Since the multi-fidelity feedback allows one to estimate an arm's true reward mean with observations of every fidelity, we pick the tightest one as arm $k$'s final confidence bounds as follows,

$$\text{UCB}_{k,t} = \min_{m \in \mathcal{M}} \text{UCB}_{k,t}^{(m)}, \quad \text{LCB}_{k,t} = \max_{m \in \mathcal{M}} \text{LCB}_{k,t}^{(m)}. \quad (5)$$

We use the above $\text{UCB}_{k,t}$ and $\text{LCB}_{k,t}$ formulas to select the two critical arms in each round and decide when to terminate the LUCB (Line 3). We present the LUCB framework in Algorithm 1. The next step is to decide fidelities for exploring both critical arms in each round ( Line 5).

### 3.2.2 Exploration Procedures

To address the accuracy-cost trade-off in fidelity selections, we devise a UCB-type policy which "finds" the optimal fidelity $m_k^*$ in Eq.(3) for each arm $k$ (EXPLORE-A) and an explore-then-commit policy stopping at a good fidelity that is at least half as good as the optimal one (EXPLORE-B).

**Notations.** The lower bound in Eq.(2) implies that there exists an optimal fidelity $m_k^*$ for exploring arm $k$. As Eq.(3) shows, the optimal fidelity $m_k^*$ maximizes the $\Delta_k^{(m)}/\sqrt{\lambda^{(m)}}$, where the $\Delta_k^{(m)}$ defined in Eq.(1) consists of two unknown reward means: $\mu_k^{(m)}$ and $\mu_1^{(M)}$ (or $\mu_2^{(M)}$). That is, calculating $\Delta_k^{(m)}$ for all $k$ needs the top two arms' reward means $\mu_1^{(M)}$ and $\mu_2^{(M)}$ which are unknown a priori. To address the issue, we assume the knowledge of an upper bound of the optimal arm's reward mean $\tilde{\mu}_1^{(M)}$ and a lower bound of the second-best arm's reward mean $\tilde{\mu}_2^{(M)}$ (see Remark 3.3 for how to obtain $\tilde{\mu}_1^{(M)}$ and $\tilde{\mu}_2^{(M)}$ in real-world applications). With the $\mu_1^{(M)}$ and $\mu_2^{(M)}$ replaced by $\tilde{\mu}_1^{(M)}$ and $\tilde{\mu}_2^{(M)}$, we define the ancillary reward gaps $\tilde{\Delta}_k^{(m)}$ and the ancillary optimal fidelity $\tilde{m}_k^*$ as follows,

$$\tilde{\Delta}_k^{(m)} := \begin{cases} \tilde{\mu}_1^{(M)} - (\mu_k^{(m)} + \zeta^{(m)}) & \forall k \neq 1 \\ (\mu_k^{(m)} - \zeta^{(m)}) - \tilde{\mu}_2^{(M)} & \forall k = 1 \end{cases}, \quad \text{and} \quad \tilde{m}_k^* := \arg\max_{m \in \mathcal{M}} \frac{\tilde{\Delta}_k^{(m)}}{\sqrt{\lambda^{(m)}}}.$$

Especially, we have $m_k^* \geqslant \tilde{m}_k^*$ because the cost $\lambda^{(m)}$ is non-decreasing and the replacement enlarges the numerator of the $\arg\max$ item in Eq.(3), and, when the bounds $\tilde{\mu}_1^{(M)}$ and $\tilde{\mu}_2^{(M)}$ are close the the true reward means, we have $m_k^* = \tilde{m}_k^*$; hence, as $\Delta_k^{(m_k^*)} > 0$, we assume $\Delta_k^{(\tilde{m}_k^*)} > 0$ for all arms $k$ as well. To present the next two procedures, we define the estimate of $\tilde{\Delta}_k^{(m)}$ as follows,

$$\hat{\Delta}_{k,t}^{(m)} := \begin{cases} \tilde{\mu}_1^{(M)} - (\hat{\mu}_{k,t}^{(m)} + \zeta^{(m)}) & \forall k \neq \ell_t \\ (\hat{\mu}_{k,t}^{(m)} - \zeta^{(m)}) - \tilde{\mu}_2^{(M)} & \forall k = \ell_t \end{cases},$$

where the $\ell_t$ is the estimated optimal arm by LUCB. For simplify, we omit the input $\ell_t$ for $\hat{\Delta}_{k,t}^{(m)}(\ell_t)$ in the LHS of this definition and thereafter.

**EXPLORE-A**  We devise the $\Delta_k^{(m)}/\sqrt{\lambda^{(m)}}$'s upper confidence bounds (f-UCB) as follows, for any fidelity $m \in \mathcal{M}$ and arms $k \in \mathcal{K}$, $\text{f-UCB}_{k,t}^{(m)} := \hat{\Delta}_{k,t}^{(m)}/\sqrt{\lambda^{(m)}} + \sqrt{2\log N_{k,t}/(\lambda^{(m)} N_{k,t}^{(m)})}$, where the $N_{k,t} := \sum_{m \in \mathcal{M}} N_{k,t}^{(m)}$ is the total number of times of pulling arm $k$ up to time $t$. Whenever the arm $k$ is selected by LUCB, we pick the fidelity $m$ that maximizes its $\text{f-UCB}_{k,t}^{(m)}$ to explore it (see Line 2 in Algorithm 2). For any arm $k$, this policy guarantees that most of the arm's pulling are on its estimated optimal fidelity $\tilde{m}_k^*$, or formally, $N_{k,t}^{(m)} = O(\log(N_{k,t}^{(\tilde{m}_k^*)}))$ for any fidelity $m \neq \tilde{m}_k^*$ as Lemma E.2 in Appendix shows. Therefore, EXPLORE-A spends most cost on the fidelity $\tilde{m}_k^*$.

**EXPLORE-B**  The cost of finding the estimated optimal fidelity $\tilde{m}_k^*$ in EXPLORE-A can be large. To avoid this cost, we devise another approach that stops exploration when finding a *good* fidelity $\hat{m}_k^*$. That is, instead of finding the $\tilde{m}_k^*$ that maximizes $\tilde{\Delta}_k^{(m)}/\sqrt{\lambda^{(m)}}$, we stop at a fidelity $\hat{m}_k^*$ whose $\tilde{\Delta}_k^{(m)}/\sqrt{\lambda^{(m)}}$ is at least half as large as that of $\tilde{m}_k^*$, i.e.,

$$\frac{\tilde{\Delta}_k^{(\hat{m}_k^*)}}{\sqrt{\lambda^{(\hat{m}_k^*)}}} \geqslant \frac{1}{2} \cdot \frac{\tilde{\Delta}_k^{(\tilde{m}_k^*)}}{\sqrt{\lambda^{(\tilde{m}_k^*)}}}. \tag{6}$$

We prove that the above inequality holds for $\hat{m}_k^* = \arg\max_{m \in \mathcal{M}} \hat{\Delta}_{k,t}^{(m)}/\sqrt{\lambda^{(m)}}$ when the condition in Line 10 of Algorithm 2 holds (see Lemma E.3). Hence, for each arm $k$, EXPLORE-B explores it at all fidelities uniformly, and, when the condition in Line 10 holds, EXPLORE-B finds a good fidelity $\hat{m}_k^*$ and keeps choosing $\hat{m}_k^*$ for exploring arm $k$ since then.

**Remark 3.3** (On the bounds $\tilde{\mu}_1^{(M)}$ and $\tilde{\mu}_2^{(M)}$ utilized in Algorithm 2 in the hyperparameter optimization application)**.** Although the exact reward means are typically not accessible, it is easy to get a good approximation of them satisfying our requirements based on domain knowledge. For example,

---

**Algorithm 2** EXPLORE Procedures

---

1: **procedure** EXPLORE-A($k$)
2:  $\quad m_{k,t} \leftarrow \arg\max_{m\in\mathcal{M}} \texttt{f-UCB}_{k,t}^{(m)}$
3:  $\quad$ Pull $(k, m_{k,t})$, observe reward, and update corresponding statistics
4: **procedure** EXPLORE-B($k$)
5:  $\quad$ **if** $\texttt{isFixed}_k$ **then** $\qquad\qquad\qquad\qquad\qquad\qquad \triangleright \texttt{isFixed}_k$ is initialized as $\texttt{False}$
6:  $\quad\quad$ Pull $(k, \hat{m}_k^*)$, observe reward, and update corresponding statistics
7:  $\quad$ **else**
8:  $\quad\quad$ **for** each fidelity $m$ **do**
9:  $\quad\quad\quad$ Pull $(k, m)$, observe reward, and update corresponding statistics
10: $\quad\quad$ **if** $\max_{m\in\mathcal{M}} \frac{\hat{\Delta}_{k,t}^{(m)}}{\sqrt{\lambda^{(m)}}} > 3\sqrt{\frac{\log(L/\delta)}{\lambda^{(1)} N_{k,t}^{(m)}}}$ **then**
11: $\quad\quad\quad$ $\hat{m}_k^* \leftarrow \arg\max_{m\in\mathcal{M}} \frac{\hat{\Delta}_{k,t}^{(m)}}{\sqrt{\lambda^{(m)}}}$ and $\texttt{isFixed}_k \leftarrow \texttt{True}$

---

in an image classification task, an easy approximation of the best arm's reward means is to use the reward from a perfect classification, i.e., $\tilde{\mu}_1^{(M)} = 1.0$, which clearly satisfies $\tilde{\mu}_1^{(M)} \geq \mu_1^{(M)}$. For the reward of the second-best arm, we can use the performance of a commonly used model that has a fairly good performance based on benchmarked results as a good approximation. One can easily find the benchmarked performance of commonly used models on a wide range of well-defined machine learning tasks on the `Papers with code` website.[2] For novel tasks without well-benchmarked results, one pragmatic way to get $\tilde{\mu}_2^{(M)}$ is to use the result from a particular default machine learning model without any tuning.

### 3.3 Cost Complexity Upper Bound Analysis

In the following, we present the cost complexity upper bounds of Algorithm 1 with above two procedures in Theorem 3.4 respectively.

**Theorem 3.4** (Cost complexity upper bounds for Algorithm 1 with procedure in Algorithm 2)**.** *Given $L \geqslant 4KM$, Algorithm 1 outputs the optimal arm with a probability at least $1 - \delta$. The cost complexities of Algorithm 1 with different fidelity selection procedures in Algorithm 2 are upper bounded as follows,*

$$(\text{EXPLORE-A}) \qquad \mathbb{E}[\Lambda] = O\left(\tilde{H}\log\left(\frac{L(\tilde{H}+\tilde{G})}{\lambda^{(1)}\delta}\right) + \tilde{G}\log\log\left(\frac{L(\tilde{H}+\tilde{G})}{\lambda^{(1)}\delta}\right)\right), \qquad (7)$$

$$(\text{EXPLORE-B}) \qquad \mathbb{E}[\Lambda] = O\left(\tilde{H}\sum_{m\in\mathcal{M}}\frac{\lambda^{(m)}}{\lambda^{(1)}}\log\left(\frac{\sum_{m\in\mathcal{M}}(\lambda^{(m)}/\lambda^{(1)})\tilde{H}L}{\lambda^{(1)}\delta}\right)\right), \qquad (8)$$

*where* $\tilde{H} := \sum_{k\in\mathcal{K}}\frac{\lambda^{(\tilde{m}_k^*)}}{(\tilde{\Delta}_k^{(\tilde{m}_k^*)})^2}$, *and* $\tilde{G} := \sum_{k\in\mathcal{K}}\sum_{m\neq\tilde{m}_k^*}\left(\frac{\tilde{\Delta}_k^{(\tilde{m}_k^*)}}{\sqrt{\lambda^{(\tilde{m}_k^*)}}} - \frac{\tilde{\Delta}_k^{(m)}}{\sqrt{\lambda^{(m)}}}\right)^{-2}$.

**EXPLORE-A *vs.* EXPLORE-B** When $\tilde{G} = O(M\tilde{H})$, the cost complexity upper bound of EXPLORE-A is less than that of EXPLORE-B (see Figure 1a). However, when $\tilde{G}$ is far more larger than $M\tilde{H}$, EXPLORE-B is better (see Figure 1b). For example, when there is a fidelity $m' (\neq \tilde{m}_k^*)$ whose $\Delta_k^{(m')}/\sqrt{\lambda^{(m')}}$ is very close to that of fidelity $\tilde{m}_k^*$ (the case in Figure 1b), this $\tilde{G}$ would be very large

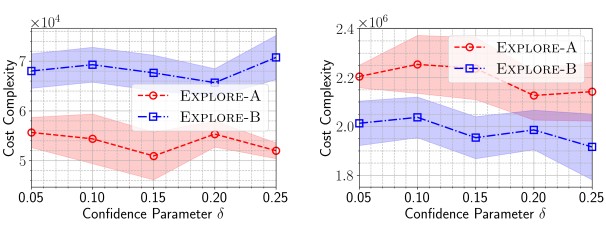

(a) EXPLORE-A is better  $\qquad$ (b) EXPLORE-B is better

Figure 1: EXPLORE-A vs. EXPLORE-B

[2]https://paperswithcode.com/sota

because EXPLORE-A needs to pay a high cost to distinguish fidelity $m'$ from $\tilde{m}_k^*$; while in this scenario, EXPLORE-B stops by either $m'_k$ or $\tilde{m}_k^*$ since their $\tilde{\Delta}_k^{(m)}/\sqrt{\lambda^{(m)}}$ are similar and, therefore, enjoys a smaller cost complexity upper bound. We report the numerical comparisons between both procedures in Figure 1. The detailed setup of the simulations is given in Appendix B.

**Remark 3.5** (Tightness of cost complexity bounds). The first term of cost complexity upper bound for EXPLORE-A in Eq.(7) matches the cost complexity lower bound in Eq.(2) up to a constant when $\tilde{m}_k^* = m_k^*$ and $\tilde{H} = H$ (i.e., when $\tilde{\mu}_1^{(M)}$ and $\tilde{\mu}_2^{(M)}$ are close to their ground truth values). The cost complexity upper bound of EXPLORE-B in Eq.(8) matches the lower bound with an additional $\sum_{m \in \mathcal{M}} \lambda^{(m)}/\lambda^{(1)}$ coefficient when $\tilde{m}_k^* = m_k^*$ and $\tilde{H} = H$.

**Remark 3.6** (Comparison to classic MAB's sample complexity). If we reduce our cost complexity upper bound result in MF-MAB to classic (single-fidelity) MAB, i.e., letting $M = 1, \lambda^{(m)} = 1$, then both cost complexity upper bounds reduce to $O(\sum_k (1/\Delta_k^2) \log(1/\delta))$ where $\Delta_k := \mu_1 - \mu_k$, which is exactly the classic sample complexity upper bound for (single-fidelity) BAI [27, 21].

## 4 Regret Minimization

In this section, we study the regret minimization objective: given a budget $\Lambda \in \mathbb{R}^+$, minimize the regret—the cumulative difference between the optimal policy's rewards and an algorithm's. We define the reward obtained in each time slot as the pulled arm's true reward mean (realized at the highest fidelity, but *unrevealed* to the learner), no matter at which fidelity the arm is pulled, while the learner's observation depends on the pulled fidelity as Section 2 shows. Under this reward definition, the optimal policy is to constantly pull the optimal arm 1 with the lowest fidelity $m = 1$. Consequently, the expected regret can be expressed as follows,

$$\mathbb{E}[R(\Lambda)] := \frac{\Lambda}{\lambda^{(1)}} \mu_1^{(M)} - \mathbb{E}\left[\sum_{t=1}^{N} \mu_{I_t}^{(M)}\right], \tag{9}$$

where $N := \max\{n : \sum_{t=1}^{n} \lambda^{(m_t)} \leqslant \Lambda\}$ is the total number of time slots, and $I_t$ is the arm pulled by a concerned algorithm at time slot $t$. Next, we illustrate the regret definition's real-world applications in Remark 4.1.

**Remark 4.1** (Applications of the new regret definition). One typical application of the regret minimization problem under MF-MAB is a variant of the advertisement distribution problem [13, 10]. In this problem, the objective is to maximize the total return from all the distributed ads within a fixed marketing budget (e.g., in terms of money). We have the following mapping between the application-specific concepts and concepts in MF-MAB. **Arm:** The ads to distribute are the arms. **Reward:** The return from each of the ads, once distributed, is the corresponding ground-truth reward $\mu_k^{(M)}$. **Low fidelity:** A minimum cost is needed every time any ad is distributed. For example, the minimum cost may include the necessary resource needed to ensure that the ad satisfies legal and regulatory requirements and to distribute the ad on the designated platform. This minimum cost can be considered the lowest fidelity cost, which can never be waived. **High fidelity:** since the expected return from different ads can be vastly different, one needs a good estimation on the expected return so as to select the profitable ads to distribute. The cost needed to get a reliable estimate of the expected return can be considered the highest fidelity cost. For example, doing a large-scale user study/survey, and/or consulting experts can give a good estimate of the expected return from the ads, which, however, is resource-consuming.

This type of application is also common in production management with uncertainty where without knowing the expected return of the concerned products, the decision maker faces the two options of (1) spending the minimum resource needed to directly produce certain products; and (2) spending more resource to first get a good estimates of the expected returns from the different options and then put the ones with the highest expected returns into production.

**Remark 4.2** (Comparison to regret definition of Kandasamy et al. [19]). Kandasamy et al. [19] defined the per time slot reward as the pulled arms' true reward mean multiplied by the cost, i.e., $\lambda^{(m_t)} \mu_{I_t}^{(M)}$, and defined their regret as, $\mathbb{E}[R'(\Lambda)] := \Lambda \mu_1^{(M)} - \mathbb{E}[\sum_{t=1}^{N} \lambda^{(m_t)} \mu_{I_t}^{(M)}]$. We note that multiplying the reward mean with the fidelity-level cost, $\lambda^{(m_t)} \mu_{I_t}^{(M)}$, does not fit into the applications

in Remark 4.1, and thus we provide an alternative definition in Eq.(9) to fit our needs. Comparing the formula of both regret definitions, we have $\mathbb{E}[R'(\Lambda)] \leqslant \lambda^{(1)}\mathbb{E}[R(\Lambda)]$. Note that both regret definitions are very different, so as their bound analysis and algorithm design.

We first present both the problem-independent (worst-case) and problem-dependent regret lower bounds in Section 4.1 and then devise an elimination algorithm whose worst-case upper bounds match the worst-case lower bound up to some logarithmic factors and whose problem-dependent upper bound matches the problem-dependent lower bound in a class of MF-MAB in Section 4.2.

## 4.1 Regret Lower Bound

We present the problem-independent regret lower bound in Theorem 4.3 and the problem-dependent regret lower bound in Theorem 4.4. Both proofs are deferred to Appendix F.1 and F.2 respectively.

**Theorem 4.3** (Problem-independent regret lower bound). *Given budget $\Lambda$, the regret of MF-MAB is lower bounded as follows,*

$$\inf_{Algo} \sup_{\mathcal{I}} \mathbb{E}\left[R(\Lambda)\right] \geqslant \Omega\left(K^{1/3}\Lambda^{2/3}\right),$$

*where the* $\inf$ *is over any algorithms, the* $\sup$ *is over any possible* MF-MAB *instances* $\mathcal{I}$.

**Theorem 4.4** (Problem-dependent lower bound). *For any consistent policy that, after spending $\Lambda$ budgets, fulfills that for any suboptimal arm $k$ (with $\Delta_k^{(M)} > 0$) and any $a > 0$, $\mathbb{E}[N_k^{(\forall m)}(\Lambda)] = o(\Lambda^a)$, its regret is lower bounded by the following inequality,*

$$\liminf_{\Lambda \to \infty} \frac{\mathbb{E}\left[R(\Lambda)\right]}{\log(\Lambda)} \geqslant \sum_{k \in \mathcal{K}} \min_{m:\Delta_k^{(m)}>0} \left(\frac{\lambda^{(m)}}{\lambda^{(1)}}\mu_1^{(M)} - \mu_k^{(M)}\right) \frac{C}{(\Delta_k^{(m)})^2}.$$

## 4.2 An Elimination Algorithm and Its Regret Upper Bound

In this section, we propose an elimination algorithm for MF-MAB based on Auer and Ortner [1]. This algorithm proceeds in phases $p = 0, 1, \ldots$ and maintains a candidate arm set $\mathcal{C}_p$. The set $\mathcal{C}_p$ is initialized as the full arm set $\mathcal{K}$ and the algorithm gradually eliminates arms from the set until there is only one arm remaining. When the candidate arm set contains more than one arms, the algorithm explores arms with the highest fidelity $M$, and when the set $|\mathcal{C}_p| = 1$, the algorithm exploits the singleton in the set with the lowest fidelity $m = 1$. We present the detail in Algorithm 3.

---

**Algorithm 3** Elimination for MF-MAB

---

1: **Input:** full arm set $\mathcal{K}$, budget $\Lambda$, and parameter $\varepsilon$
2: **Initialization:** phase $p \leftarrow 0$, candidate set $\mathcal{C}_p \leftarrow \mathcal{K}$
3: **while** $p < \log_2 \frac{2}{\varepsilon}$ and $|\mathcal{C}_p| > 1$ **do**
4:      pull each arm $k \in \mathcal{C}_p$ in highest fidelity $M$ such that $T_k^{(M)} = \left\lceil 2^{2p} \log \frac{\Lambda}{2^{2p}\lambda^{(M)}} \right\rceil$
5:      Update reward means $\hat{\mu}_{k,p}^{(M)}$ for all arms $k \in \mathcal{C}_p$
6:      $\mathcal{C}_{p+1} \leftarrow \{k \in \mathcal{C}_p : \hat{\mu}_{k,p}^{(M)} + 2^{-p+1} > \max_{k' \in \mathcal{C}_p} \hat{\mu}_{k',p}^{(M)}\}$             ▷ Elimination
7:      $p \leftarrow p + 1$
8: Pull the remaining arms of $\mathcal{C}_p$ in turn in fidelity $m = 1$ until the budget runs up

---

**Remark 4.5** (Real-world implication of the 2-stage algorithm design). There are real-world applications, e.g., the advertisement distribution problem in Remark 4.1, where the explorations are conducted at the high fidelity (e.g., a large-scale user study) and the exploitations are conducted at the low fidelity (e.g., advertisement distribution via some platforms). This corroborates our algorithm design which also explores at high fidelity and exploits at low fidelity. On the other hand, the fact that our algorithm enjoys the tight regret performance comparing to regret lower bound (both problem-independent and -dependent, see Remarks 4.9 and 4.7) also implies that the approach of first conducting large-scale user study and then massive distributing good ads used in real-world advertisement distribution is reasonable.

### 4.2.1 Analysis Results

We first present the problem-dependent regret upper bound of Algorithm 3 in Theorem 4.6. Its full proof is deferred to Appendix F.3.

**Theorem 4.6** (Problem-Dependent Regret Upper Bound for Algorithm 3). *For any $\varepsilon > 0$. Algorithm 3's regret is upper bounded as follows,*

$$
\mathbb{E}[R(\Lambda)] \leqslant \max_{k:\Delta_k^{(M)} \leqslant \varepsilon} \frac{\Lambda}{\lambda^{(1)}} \Delta_k^{(M)}
$$

$$
+ \sum_{k:\Delta_k^{(M)} > \varepsilon} \left( \left( \frac{\lambda^{(M)}}{\lambda^{(1)}} \mu_1^{(M)} - \mu_k^{(M)} \right) \left( \frac{16}{(\Delta_k^{(M)})^2} \log \frac{\Lambda (\Delta_k^{(M)})^2}{16\lambda^{(M)}} + \frac{48}{(\Delta_k^{(M)})^2} + 1 \right) + \frac{64}{\Delta_k^{(M)}} \right) \quad (10)
$$

$$
+ \sum_{k:\Delta_k^{(M)} \leqslant \varepsilon} \left( \left( \frac{\lambda^{(M)}}{\lambda^{(1)}} \mu_1^{(M)} - \mu_k^{(M)} \right) \left( \frac{16}{\varepsilon^2} \log \left( \frac{\Lambda \varepsilon^2}{16\lambda^{(M)}} \right) + \frac{32}{3\varepsilon^2} + 1 \right) + \frac{64}{\varepsilon} . \right)
$$

*Especially, if letting $\varepsilon$ go to zero and budget $\Lambda$ go to infinity, the above upper bound becomes*

$$
\limsup_{\Lambda \to \infty} \frac{\mathbb{E}[R(\Lambda)]}{\log(\Lambda)} \leqslant \sum_{k \in \mathcal{K}} \left( \frac{\lambda^{(M)}}{\lambda^{(1)}} \mu_1^{(M)} - \mu_k^{(M)} \right) \frac{16}{(\Delta_k^{(M)})^2} . \quad (11)
$$

**Remark 4.7** (Tightness of problem-dependent regret bounds). The problem-dependent regret bounds are tight for a class of MF-MAB instances. For instances fulfilling the condition that $\{m : \Delta_k^{(m)} > 0\} = \{M\}$ for all arm $k \in \mathcal{K}$, then the problem-dependent regret lower bound matches the upper bound. This kind of instances covers a vast number of real world scenarios. Because in practice, the highest fidelity $M$ is often defined as the only fidelity where the optimal arm can be distinguished from the suboptimal arms. For example, in neural architecture search, the process of increasing the training sample size (fidelity) stops when one architecture performs much better than others.

Letting $\varepsilon = (K \log \Lambda / \Lambda)^{1/3}$ in Eq.(10) of Theorem 4.6, one can obtain a problem-independent regret upper bound of Algorithm 3 in Theorem 4.8 as follows.

**Theorem 4.8** (Problem-Independent Regret Upper Bound for Algorithm 3). *Letting $\varepsilon = (K \log \Lambda / \Lambda)^{1/3}$, Algorithm 3's regret is upper bounded as follows,*

$$
\mathbb{E}[R(\Lambda)] \leqslant O \left( K^{1/3} \Lambda^{2/3} (\log \Lambda)^{1/3} \right) .
$$

**Remark 4.9** (Tightness of problem-independent regret bounds). The problem-independent regret upper bound matches the problem-independent lower bound in terms of the number of arms $K$ and up to some logarithmic factor in terms of the budget $\Lambda$.

## 5 Future Directions

We note that the BAI's exploration procedures in Algorithm 2 require some prior knowledge of the top two arms' reward mean estimates $\tilde{\mu}_1^{(M)}$ and $\tilde{\mu}_2^{(M)}$ as input. Although such prior knowledge is easy to access in many real world applications (see Remark 3.3), this is not a common assumption in bandits literature. Without relying on this prior knowledge, we devise a third procedure EXPLORE-C in Appendix D which starts from the lower fidelity and gradually increase fidelity when necessary. However, its cost complexity upper bound is incomparable to the lower bound in Eq.(2) and can be very large. Therefore, one interesting future direction is to devise BAI algorithms without this prior knowledge but still enjoying good theoretical performance.

Another interesting future direction is to quantify the cost of identifying optimal fidelities and improve the current cost complexity lower bound in Eq.(2). Note that the second term of cost complexity upper bound for EXPLORE-A in Eq.(7) has no correspondence in the lower bound, so does the additional factor $\sum_{m \in \mathcal{M}} \lambda^{(m)} / \lambda^{(1)}$ of the bound for EXPLORE-B in Eq.(8). These two additional terms may correspond to the cost of finding the optimal fidelity $m_k^*$ which is not accounted in the lower bound in Theorem 3.1.

## Acknowledgements

We thank Quanlu Zhang for his insightful discussion on the potential applications of `MF-MAB`, especially, neural architecture search. The work of John C.S. Lui and Xuchuang Wang was supported in part by the RGC's GRF 14207721 and SRFS2122-4S02.

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

## Supplementary Materials

## A    Additional Remarks on Analysis

### A.1    Remarks on Best Arm Identification

**Remark A.1** (On choice among EXPLORE-A and EXPLORE-B). For when do we prefer EXPLORE-A than EXPLORE-B, one can informally say if $\tilde{G}$ is small ($\tilde{G} = O(M\tilde{H})$), EXPLORE-A is preferred, and, otherwise if $\tilde{G}$ is large, EXPLORE-B is preferred. Recall that

$$\tilde{G} := \sum_{k \in \mathcal{K}} \sum_{m \neq \tilde{m}_k^*} \left( \frac{\tilde{\Delta}_k^{(\tilde{m}_k^*)}}{\sqrt{\lambda^{(\tilde{m}_k^*)}}} - \frac{\tilde{\Delta}_k^{(m)}}{\sqrt{\lambda^{(m)}}} \right)^{-2}.$$

To give an intuition of when $\tilde{G}$ is large or small, let us neglect the tilde for simplicity and fix a suboptimal arm $k$. The term $\frac{\Delta_k^{(m)}}{\sqrt{\lambda^{(m)}}}$ corresponds to the effectiveness of pulling arm $k$ at fidelity $m$ for distinguishing suboptimal arm $k$. As EXPLORE-A aims to find out the optimal (most effective) fidelity $m_k^*$ for distinguishing suboptimal arm $k$, it needs to compare the effectiveness of different fidelities, that is, their *effectiveness gap* $\frac{\Delta_k^{(m_k^*)}}{\sqrt{\lambda^{(m_k^*)}}} - \frac{\Delta_k^{(m)}}{\sqrt{\lambda^{(m)}}}$, which is similar to bandit's reward gap (reward mean difference) $\mu^* - \mu_k$. Therefore, for this suboptimal arm $k$, if all its *effectiveness gaps* are large, that is, the effectiveness of its optimal fidelity is much larger than other suboptimal fidelities, then the contribution of this arm $k$ to the quantity $G$ would be small, and *vice versa*. If for all arms $k \in \mathcal{K}$, all of their effectiveness gaps are large, then the $G$ would be small; otherwise, $G$ can be large.

For a more sensible example, one can check the simulation set up (deferred to Appendix A) of the empirical comparison of EXPLORE-A and EXPLORE-B in Figure 1. In Figure 1a's set up, the effectiveness gaps are large and thus $G$ is small, hence EXPLORE-A outperforms EXPLORE-B, while in Figure 1b's set up, the effectiveness gaps are small and thus $G$ is large, then EXPLORE-B outperforms EXPLORE-A.

Practically, one can just run both EXPLORE-A and -B in parallel for each arm $k$. If EXPLORE-A starts to converge on a fidelity, i.e., a large proportion of pull times on this arm is at this fidelity, we stop EXPLORE-B and continue EXPLORE-A. Otherwise, if EXPLORE-B commits to a fidelity at first, we stop EXPLORE-A and follow EXPLORE-B to commit this fidelity. With this heuristic approach, we can expect the empirical cost is close to the smaller one of running either EXPLORE-A or EXPLORE-B alone.

**Remark A.2** (On the choice of "half" in EXPLORE-B as good fidelity). The selection of "half" is just for the simplicity of presentation, and any constant between $(0, 1)$ would work. If a constant $C \in (0, 1)$ is used in Eq.(6), then the factor 3 of right hand side in the condition of Line 10 of EXPLORE-B becomes $\frac{1+C}{1-C}$, and there would be an additional $(C^{-2} + (1-C)^{-2})$ multiplier appearing in the cost complexity upper bound of EXPLORE-B in Eq.(8). In the $(C^{-2} + (1-C)^{-2})$ multiplier, the first term $C^{-2}$ factor is because a small $C$ implies that EXPLORE-B may commit to a very inefficient suboptimal fidelity and thus suffer a high cost, while the second term $(1-C)^{-2}$ corresponds to that if $C$ is close to 1, then it becomes very difficult for EXPLORE-B to find a good fidelity to commit, and thus EXPLORE-B has to spend a large cost in exploration.

### A.2    Remarks on Regret Minimization

**Remark A.3** (Relation to partial monitoring). Recall that, under our regret definition in Eq.(9), the optimal policy is to pull the optimal arm 1 at the lowest fidelity, and any exploration at the highest fidelity results in nonzero regret. This clear separation of exploration and exploitation is similar to the hard case of partial monitoring [2, 23] where the decision maker needs to choose the globally observable actions with nonzero regret cost to do exploration and then, when exploiting, choose locally observable actions. The regret of the hard case of partial monitoring is $\Theta(T^{2/3})$ where $T$ is the decision round, which is also similar to our $\tilde{O}(\Lambda^{2/3})$ regret bound.

# B  Simulation Details of Figure 1

In Figure 1, we report the cost complexities of Algorithm 1 with EXPLORE-A and Algorithm 1 with EXPLORE-B (let $\tilde{\mu}_1^{(M)} = 0.95$ and $\tilde{\mu}_2^{(M)} = 0.75$). We set the confidence parameter $\delta$ as $0.05, 0.1, 0.15, 0.2, 0.25$ respectively in comparing the performance of both procedures. For each simulation, we run 100 trials, plot their cost complexities' mean as markers and their deviation as shaded regions. We present the parameters of MF-MAB instances of Figures 1a and 1b in Tables 1 and 2 respectively. We note that it is more difficult to find the optimal fidelity $m_k^*$ in the MF-MAB instances for Figure 1b because (1) there are more fidelities choices in this instance than that of Figure 1a; (2) the value of $\tilde{\Delta}_k^{(m)}/\sqrt{\lambda^{(m)}}$ are closer in the second instance than that of Figure 1a.

Table 1: Figure 1a's MF-MAB with $K = 5$ arms and $M = 3$ fidelities

| Parameters | $\mu_1^{(m)}$ | $\mu_2^{(m)}$ | $\mu_3^{(m)}$ | $\mu_4^{(m)}$ | $\mu_5^{(m)}$ | $\zeta^{(m)}$ | $\lambda^{(m)}$ |
|---|---|---|---|---|---|---|---|
| $m = 1$ | 0.70 | 0.75 | 0.50 | 0.50 | 0.30 | 0.30 | 1 |
| $m = 2$ | 0.80 | 0.775 | 0.60 | 0.55 | 0.45 | 0.15 | 1.1 |
| $m = 3$ | 0.90 | 0.80 | 0.70 | 0.60 | 0.50 | 0 | 1.2 |

Table 2: Figure 1b's MF-MAB with $K = 5$ arms and $M = 5$ fidelities

| Parameters | $\mu_1^{(m)}$ | $\mu_2^{(m)}$ | $\mu_3^{(m)}$ | $\mu_4^{(m)}$ | $\mu_5^{(m)}$ | $\zeta^{(m)}$ | $\lambda^{(m)}$ |
|---|---|---|---|---|---|---|---|
| $m = 1$ | 0.83 | 0.82 | 0.76 | 0.82 | 0.70 | 0.10 | 1 |
| $m = 2$ | 0.84 | 0.83 | 0.80 | 0.80 | 0.72 | 0.08 | 1.1 |
| $m = 3$ | 0.85 | 0.85 | 0.80 | 0.82 | 0.74 | 0.06 | 1.2 |
| $m = 4$ | 0.85 | 0.86 | 0.80 | 0.80 | 0.76 | 0.04 | 1.3 |
| $m = 5$ | 0.90 | 0.88 | 0.86 | 0.84 | 0.80 | 0 | 1.4 |

# C  Additional Experiments

In this section, we present an additional numerical simulation to compare our two exploration policies, Explore-A and Explore-B, with a naive baseline that is always selecting the highest fidelity to explore arms. All three policies are employed under the LUCB algorithmic framework (Algorithm 1). This simulation aims to illustrate when our exploration policies Explore-A and Explore-B are better than the naive baseline.

**Experiment set up.** We consider a MF-MAB model consisting of $K = 5$ arms and $M = 5$ fidelities. The reward means are set as $\mu_k^{(m)} = 0.1(k + 4) + 0.01(m - 5)$, where, for example, $\mu_5^{(5)} = 0.9$ is the true reward mean of optimal arm and its low-fidelity reward means are $\{0.89, 0.88, 0.87, 0.86\}$ accordingly, etc. The error upper bound in different fidelities $\zeta^{(m)}$ are $\{0.04, 0.03, 0.02, 0.01, 0\}$. We set the cost of fidelity $m \in \{1, 2, 3, 4\}$ as $\lambda^{(m)} = m$ respectively, and consider four different cases $I \in \{1, 2, 3, 4\}$ of the cost of highest fidelity $\lambda^{(5)} = 5I$. That is, from Case 1 to Case 4, the only difference is that the cost of highest fidelity increases.

We set the confidence parameter $\delta = 0.1$ and run each experiment for 100 trials. Averaged over these 100 trials of each experiment, we present the empirical cost complexity in the table as follows,

Table 3: Cost complexity under four different cases

| Cost Complexity ($\times 10^4$) | Case 1 | Case 2 | Case 3 | Case 4 |
|---|---|---|---|---|
| Explore-A | 20.5 | 20.0 | 17.4 | 17.4 |
| Explore-B | 36.1 | 47.9 | 59.5 | 69.4 |
| Baseline | 21.4 | 42.0 | 64.6 | 86.2 |

**Discussion of additional simulation result.** In all four cases, the cost complexity of Explore-A does not change too much. Especially, when the cost of highest fidelity becomes larger, it becomes

easier for `Explore-A` to find out the optimal fidelity (e.g., the highest fidelity is obviously not the optimal one) and hence enjoy a lower cost complexity. The cost complexity of both `Explore-B` and `Baseline` increases over these four cases.

When the cost of highest fidelity is expensive, e.g., Cases 3 and 4, both `Explore-A` and `Explore-B` outperform `Baseline`. This highlights the advantage of utilizing low fidelity to explore arms when the cost of high fidelity is large. When the cost of highest fidelity is cheap, e.g., Case 1, the cost complexity of `Explore-A` is similar to `Baseline`, while `Explore-B` is worse than `Baseline`. This suggests that when the cost of highest fidelity is similar to that of lower fidelity, the `Baseline` may be preferred.

In this comparison, `Explore-A` always outperforms `Explore-B`. For another scenario where `Explore-B` is better than `Explore-B`, please refer to Figure 1b.

## D  A Third Fidelity Selection Procedure: EXPLORE-C

Besides the EXPLORE-A and -B procedures, here we consider a third naïve and conservative idea for fidelity selection that one should start from low risk (cost), gradually increase the risk (cost) as the learning task needs, and stop when finding the optimal arm. To decide when to increase the fidelity for exploring an arm $k$, we use the arm's confidence radius $\beta(N_{k,t}^{(m)}, t, \delta)$ at fidelity $m$ as a measure of the amount of information left in this fidelity, and when the fidelity $m$'s confidence radius is less than the error upper bound $\zeta^{(m)}$ at this fidelity, we increase the fidelity by 1 for higher accuracy, or formally, the fidelity is selected as follows,

$$m_{k,t} \leftarrow \min \left\{ m \,\middle|\, \beta(N_{k,t}^{(m)}, t, \delta) \geqslant \zeta^{(m)} \right\}.$$

---
**Algorithm 4** EXPLORE-C Procedures

**procedure** EXPLORE-C($k$)

$\quad m_{k,t} \leftarrow \min \left\{ m \,\middle|\, \beta(N_{k,t}^{(m)}, t, \delta) \geqslant \zeta^{(m)} \right\}$

$\quad$ Pull $(k, m_{k,t})$, observe reward, and update corresponding statistics

---

**Theorem D.1** (Cost complexity upper bounds for Algorithm 1 with EXPLORE-C). *Given $L \geqslant 4KM$, Algorithm 1 outputs the optimal arm with a probability at least $1 - \delta$. The cost complexity of Algorithm 1 with EXPLORE-C are upper bounded as follows,*

$$\mathbb{E}[\Lambda] = O\left( H^{\ddagger} \log\left( \frac{L(H^{\ddagger} + Q)}{\lambda^{(1)}\delta} \right) + Q \log\left( \frac{L(H^{\ddagger} + Q)}{\lambda^{(1)}\delta} \right) \right), \tag{12}$$

*where, letting $m_k^{\ddagger}$ denote the smallest fidelity for arm $k$ such that $\Delta_k^{(m)} > 2\zeta^{(m)}$, or formally,*

$$m_k^{\ddagger} := \min\{m : \Delta_k^{(m)} > 2\zeta^{(m)}\},$$

*and we denote*

$$H^{\ddagger} := \sum_{k \in \mathcal{K}} \frac{\lambda^{(m_k^{\ddagger})}}{(\Delta_k^{(m_k^{\ddagger})})^2}, \quad Q := \sum_{k \in \mathcal{K}} \sum_{m=1}^{m_k^{\ddagger}-1} \frac{\lambda^{(m)}}{(\zeta^{(m)})^2}.$$

**Remark D.2** (EXPLORE-C *vs.* EXPLORE-A and -B). The EXPLORE-C procedure does not require additional knowledge as the other two. It is a one-size-fits-all option. If with some addition information of a specific scenario, e.g., the exact or approximated reward means of top two arms, one can use the EXPLORE-A or B.

## E  Proofs for Best Arm Identification with Fixed Confidence

### E.1  Proof of Theorem 3.1

**Lemma E.1** (Kaufmann et al. [21], Lemma 1). *Let $\nu$ and $\nu'$ be two bandit models with $K$ arms such that for all $k$, the distributions $\nu_k$ and $\nu'_k$ are mutually absolutely continuous. For any almost-surely*

*finite stopping time $\sigma$ with respect to the filtration $\{\mathcal{F}_t\}_{t\in\mathbb{N}}$ where $\mathcal{F}_t = \sigma(I_1, X_1, \ldots, I_t, X_t)$,*

$$\sum_{k=1}^{K} \mathbb{E}_\nu[N_k(\sigma)] \operatorname{KL}(\nu_k, \nu'_k) \geqslant \sup_{\mathcal{E}\in\mathcal{F}_\sigma} \operatorname{kl}(\mathbb{P}_\nu(\mathcal{E}), \mathbb{P}_{\nu'}(\mathcal{E})),$$

*where $\operatorname{kl}(x, y)$ is the binary relative entropy.*

In `MF-MAB` model, regarding each arm-fidelity $(k, m)$-pair as an individual arm, we can extend Lemma E.1 to multi-fidelity case as follows,

$$\sum_{k=1}^{K} \sum_{m=1}^{M} \mathbb{E}_\nu[N_k^{(m)}(\sigma)] \operatorname{KL}(\nu_k^{(m)}, \nu'^{(m)}_k) \geqslant \sup_{\mathcal{E}\in\mathcal{F}_\sigma} \operatorname{kl}(\mathbb{P}_\nu(\mathcal{E}), \mathbb{P}_{\nu'}(\mathcal{E})). \tag{13}$$

Next, we construct instances $\nu$ and $\nu'$. We set the reward distributions $\nu = (\nu_k^{(m)})_{(k,m)\in\mathcal{K}\times\mathcal{M}}$ as Bernoulli and the reward means fulfill $\mu_1^{(M)} > \mu_2^{(M)} \geqslant \mu_3^{(M)} \geqslant \ldots \geqslant \mu_K^{(M)}$, where $\mu_k^{(m)} = \mathbb{E}_{X\sim\nu_k^{(m)}}[X]$. We let $\nu'^{(m)}_k$ be the same to $\nu_k^{(m)}$ for all $k$ and $m$, except for that an arm $\ell \neq 1$. We set arm $\ell$'s reward means on fidelities $m \in \mathcal{M}_k$ to be $\nu'^{(m)}_\ell = \nu_1^{(M)} - \zeta^{(m)} + \epsilon$. So, in instance $\nu'$, the optimal arm is $\ell$ and its true reward mean $\mu'^{(M)}_\ell$ is slightly greater than $\mu_1^{(M)}$. This implies for the event $\mathcal{E} = \{\text{output arm 1}\}$ and any algorithm $\pi$ that can find the optimal arm with a confidence $1 - \delta$, $\mathbb{P}_{\nu,\pi}(\mathcal{E}) \geqslant 1 - \delta$ and $\mathbb{P}_{\nu',\pi}(\mathcal{E}) \leqslant \delta$. Then, from Eq.(13), we have

$$\sum_{m\in\mathcal{M}_k} \mathbb{E}_\nu[N_\ell^{(m)}(\sigma)] \operatorname{KL}(\nu_\ell^{(m)}, \nu'^{(m)}_\ell) \geqslant \sup_{\mathcal{E}\in\mathcal{F}_\sigma} \operatorname{kl}(\mathbb{P}_{\nu,\pi}(\mathcal{E}), \mathbb{P}_{\nu',\pi}(\mathcal{E}))$$

$$\geqslant \operatorname{kl}(1 - \delta, \delta)$$

$$\geqslant \log \frac{1}{2.4\delta}.$$

We rewrite the above inequality as follows,

$$\sum_{m\in\mathcal{M}_k} \lambda^{(m)} \mathbb{E}_\nu[N_\ell^{(m)}(\sigma)] \cdot \frac{\operatorname{KL}(\nu_\ell^{(m)}, \nu'^{(m)}_\ell)}{\lambda^{(m)}} \geqslant \log \frac{1}{2.4\delta}.$$

Therefore, for the arm $\ell$, our aim is to minimize its cost complexity with a constraint as follows,

$$\min_{\mathbb{E}[N_\ell^{(m)}], \forall m} \sum_{m=1}^{M} \lambda^{(m)} \mathbb{E}_\nu[N_\ell^{(m)}(\sigma)]$$

$$\text{such that } \sum_{m\in\mathcal{M}_k} \lambda^{(m)} \mathbb{E}_\nu[N_\ell^{(m)}(\sigma)] \cdot \frac{\operatorname{KL}(\nu_\ell^{(m)}, \nu'^{(m)}_\ell)}{\lambda^{(m)}} \geqslant \log \frac{1}{2.4\delta}.$$

Note that the above is a linear programming (LP) and its optimum is reached at one of its polyhedron constraint's vertex—only one $\mathbb{E}[N_\ell^{(m)}]$ is positive and all others are equal to zero.

$$\min_{\mathbb{E}[N_\ell^{(m)}], \forall m} \sum_{m=1}^{M} \lambda^{(m)} \mathbb{E}_\nu[N_\ell^{(m)}(\sigma)] \overset{(a)}{\geqslant} \min_{m\in\mathcal{M}_k} \frac{\lambda^{(m)}}{\operatorname{KL}(\nu_\ell^{(m)}, \nu'^{(m)}_\ell)} \log \frac{1}{2.4\delta}$$

$$= \min_{m\in\mathcal{M}_k} \frac{\lambda^{(m)}}{\operatorname{KL}(\nu_\ell^{(m)}, \nu_1^{(M)} - \zeta^{(m)} + \epsilon)} \log \frac{1}{2.4\delta}$$

$$\overset{(b)}{\geqslant} \min_{m\in\mathcal{M}_k} \frac{\lambda^{(m)}}{(1 + \varepsilon)\operatorname{KL}(\nu_\ell^{(m)}, \nu_1^{(M)} - \zeta^{(m)})} \log \frac{1}{2.4\delta}$$

where the inequality (a) is due to the property of LP we mentioned above, and the inequality (b) is because of the continuity of KL-divergence.

To bound the optimal arm 1's cost complexity, we use the same $\nu$ as above and construct another instance $\nu''$. The instance $\nu''$'s reward means are the same to $\nu$ except for arm 1 whose reward means

for fidelity $m \in \mathcal{M}_1$ are set as $\mu''^{(m)}_1 = \mu^{(m)}_2 + \zeta^{(m)} - \epsilon$. Then, with similar procedure as the above, we obtain

$$\min_{\mathbb{E}[N_1^{(m)}], \forall m} \sum_{m=1}^{M} \lambda^{(m)} \mathbb{E}_\nu[N_1^{(m)}(\sigma)] \geqslant \min_{m \in \mathcal{M}_1} \frac{\lambda^{(m)}}{(1+\varepsilon) \, \mathrm{KL}(\nu_1^{(m)}, \nu_2^{(M)} + \zeta^{(m)})} \log \frac{1}{2.4\delta}.$$

Summing up the above costs leads to the lower bound as follows, and letting the $\epsilon$ goes to zeros concludes the proof.

$$\mathbb{E}[\Lambda] \geqslant$$

$$\left( \min_{m \in \mathcal{M}_1} \frac{\lambda^{(m)}}{(1+\varepsilon) \, \mathrm{KL}(\nu_1^{(m)}, \nu_2^{(M)} + \zeta^{(m)})} + \sum_{k \neq 1} \min_{m \in \mathcal{M}_k} \frac{\lambda^{(m)}}{(1+\varepsilon) \, \mathrm{KL}(\nu_k^{(m)}, \nu_1^{(M)} - \zeta^{(m)})} \right) \log \frac{1}{2.4\delta}.$$

### E.2 Proof of Theorem 3.4

**Notation.** Denote the threshold $c = \frac{\mu_1^{(M)} + \max_k (\mu_k^{(m_k^*)} + \zeta^{(m_k^*)})}{2}$ as the average between the optimal arm's reward mean and the highest reward mean upper bound of other arms $k$ at fidelity $m_k^*$. Denote $\mathcal{A}_t := \{k \in \mathcal{K} : \mathrm{LCB}_{k,t} > c\}$ and $\mathcal{B}_t := \{k \in \mathcal{K} : \mathrm{UCB}_{k,t} < c\}$ as the above and below sets which respectively contain arms whose rewards are clearly higher or lower than the threshold with high probability, and let $\mathcal{C}_t := \mathcal{K} \setminus (\mathcal{A}_t \cup \mathcal{B}_t)$ as the complement of both sets' union. Then, we define two events as follows

$$\mathrm{TERM}_t := \{\mathrm{LCB}_{\ell_t,t} > \mathrm{UCB}_{u_t,t}\},$$
$$\mathrm{CROS}_t := \{\exists k \neq 1 : k \in \mathcal{A}_t\} \cup \{1 \in \mathcal{B}_t\}.$$

The $\mathrm{TERM}_t$ event corresponds to the complement of the main while loop condition in the LUCB algorithm. When the $\mathrm{TERM}_t$ event happens, the LUCB algorithm terminates. The $\mathrm{CROS}_t$ event means there exists a suboptimal arm whose $\mathrm{LCB}_{k,t}$ is greater than $c$ or that the optimal arm 1's $\mathrm{UCB}_{1,t}$ is less than $c$, both of which means that at least one arm's reward mean confidence interval incorrectly crosses the threshold $c$.

**Step 1. Prove** $\neg\mathrm{TERM}_t \cap \neg\mathrm{CROS}_t \implies (\ell_t \in \mathcal{C}_t) \cup (u_t \in \mathcal{C}_t)$. We show this statement by contradiction case by case. That is, the negation of $(\ell_t \in \mathcal{C}_t) \cup (u_t \in \mathcal{C}_t)$ cannot happen when $\neg\mathrm{TERM}_t \cap \neg\mathrm{CROS}_t$.

**Case 1:** $(\ell_t \in \mathcal{A}_t) \cap (u_t \in \mathcal{A}_t) \cap \neg\mathrm{TERM}_t$
$\implies (\ell_t \in \mathcal{A}_t) \cap (u_t \in \mathcal{A}_t) \implies |\mathcal{A}_t| \geqslant 2 \implies \exists k \neq 1 : k \in \mathcal{A}_t \implies \mathrm{CROS}_t,$
**Case 2:** $(\ell_t \in \mathcal{B}_t) \cap (u_t \in \mathcal{A}_t) \cap \neg\mathrm{TERM}_t$
$\implies \mathrm{UCB}_{\ell_t,t} < c < \mathrm{LCB}_{u_t,t} < \mathrm{UCB}_{u_t,t} \implies \emptyset \text{ (contradicts the selection of } \ell_t \text{ and } u_t\text{)},$
**Case 3:** $(\ell_t \in \mathcal{A}_t) \cap (u_t \in \mathcal{B}_t) \cap \neg\mathrm{TERM}_t$
$\implies \{\mathrm{LCB}_{\ell_t,t} > c > \mathrm{UCB}_{u_t,t}\} \cap \neg\mathrm{TERM}_t \implies \emptyset,$
**Case 4:** $(\ell_t \in \mathcal{B}_t) \cap (u_t \in \mathcal{B}_t) \cap \neg\mathrm{TERM}_t$
$\implies (\ell_t \in \mathcal{B}_t) \cap (u_t \in \mathcal{B}_t) \implies |\mathcal{B}_t| = K \implies 1 \in \mathcal{B}_t \implies \mathrm{CROS}_t.$

**Step 2. Prove $\mathbb{P}(\text{CROS}_t) \leqslant \frac{KM\delta}{Lt^3}$.** For any suboptimal arm $k \neq 1$, we bound the probability that the arm $k$ is in $\mathcal{A}_t$ as follows,

$$\mathbb{P}(k \in \mathcal{A}_t) = \mathbb{P}(\text{LCB}_{k,t} > c) = \mathbb{P}\left(\max_{m \in \mathcal{M}} \text{LCB}_{k,t}^{(m)} > c\right) \leqslant \sum_{m \in \mathcal{M}} \mathbb{P}\left(\text{LCB}_{k,t}^{(m)} > c\right)$$

$$= \sum_{m \in \mathcal{M}} \mathbb{P}\left(\hat{\mu}_{k,t}^{(m)} - \zeta^{(m)} - \beta(N_{k,t}^{(m)}, t) > c\right)$$

$$= \sum_{m \in \mathcal{M}} \mathbb{P}\left(\hat{\mu}_{k,t}^{(m)} - \mu_k^{(m)} + (\mu_k^{(m)} - \zeta^{(m)} - c) > \beta(N_{k,t}^{(m)}, t)\right)$$

$$\overset{(a)}{\leqslant} \sum_{m \in \mathcal{M}} \mathbb{P}\left(\hat{\mu}_{k,t}^{(m)} - \mu_k^{(m)} > \beta(N_{k,t}^{(m)}, t)\right) \leqslant \sum_{m \in \mathcal{M}} \sum_{n=1}^{t} \mathbb{P}(\hat{\mu}_{k,t}^{(m)} - \mu_k^{(m)} > \beta(n, t))$$

$$\leqslant \sum_{m \in \mathcal{M}} \sum_{n=1}^{t} \exp(-n(\beta(n, t))^2) = \sum_{m \in \mathcal{M}} \sum_{n=1}^{t} \frac{\delta}{Lt^4}$$

$$\leqslant \frac{M\delta}{Lt^3},$$

where the inequality (a) is due to that $\mu_k^{(m)} - \zeta^{(m)} \leqslant \mu_k^{(M)} < c$. With similar derivation, we have $\mathbb{P}(1 \in \mathcal{B}_t) \leqslant \frac{M\delta}{Lt^3}$. Noticing that $\mathbb{P}(\text{CROS}_t) \leqslant \sum_{k \neq 1} \mathbb{P}(k \in \mathcal{A}_t) + \mathbb{P}(1 \in \mathcal{B}_t)$, we have $\mathbb{P}(\text{CROS}_t) \leqslant \frac{KM\delta}{Lt^3}$.

**Step 3. Prove** $\mathbb{P}\left(\exists k \in \mathcal{K} : (N_{k,t}^{(m_k^*)} > 16N_{k,t}^*) \cap (k \in \text{Mid}_t)\right) \leqslant \frac{16\delta \sum_{k \in \mathcal{K}} \Delta_k^{-2}}{Lt^4}$, **where** $N_{k,t}^* := \frac{\log(Lt^4/\delta)}{(\Delta_k^{(\tilde{m}_k^*)})^2}$. For any fixed suboptimal arm $k \neq 1$ (with $\mu_k^{(M)} < c$), we have

$$\mathbb{P}\left((N_{k,t}^{(m_k^*)} > 16N_{k,t}^*) \cap (k \in \text{Mid}_t)\right)$$

$$= \mathbb{P}\left((N_{k,t}^{(m_k^*)} > 16N_{k,t}^*) \cap (k \notin \mathcal{A}_t \cup \mathcal{B}_t)\right)$$

$$\leqslant \mathbb{P}\left((N_{k,t}^{(m_k^*)} > 16N_{k,t}^*) \cap (\text{UCB}_{k,t} > c)\right)$$

$$= \mathbb{P}\left((N_{k,t}^{(m_k^*)} > 16N_{k,t}^*) \cap \left(\min_{m \in \mathcal{M}} \hat{\mu}_{k,t}^{(m)} + \zeta^{(m)} + \beta(N_{k,t}^{(m)}, t) > c\right)\right)$$

$$\leqslant \mathbb{P}\left((N_{k,t}^{(m_k^*)} > 16N_{k,t}^*) \cap (\hat{\mu}_{k,t}^{(m_k^*)} + \zeta^{(m_k^*)} + \beta(N_{k,t}^{(m_k^*)}, t) > c)\right)$$

$$\leqslant \mathbb{P}\left((N_{k,t}^{(m_k^*)} > 16N_{k,t}^*) \cap (\hat{\mu}_{k,t}^{(m_k^*)} - \mu_k^{(m_k^*)} > (c - \mu_k^{(m_k^*)} - \zeta^{(m_k^*)}) - \beta(N_{k,t}^{(m_k^*)}, t))\right)$$

$$\overset{(a)}{\leqslant} \mathbb{P}\left((N_{k,t}^{(m_k^*)} > 16N_{k,t}^*) \cap \left(\hat{\mu}_{k,t}^{(m_k^*)} - \mu_k^{(m_k^*)} > \frac{\Delta_k^{(\tilde{m}_k^*)}}{2} - \beta(N_{k,t}^{(\tilde{m}_k^*)}, t)\right)\right)$$

$$\overset{(b)}{\leqslant} \sum_{\tau > 16N_{k,t}^*} \mathbb{P}\left(\hat{\mu}_{k,t(\tau)}^{(m_k^*)} - \mu_k^{(m_k^*)} > \frac{\Delta_k^{(\tilde{m}_k^*)}}{4}\right)$$

$$\leqslant \sum_{\tau > 16N_{k,t}^*} \exp\left(-\frac{\tau(\Delta_k^{(\tilde{m}_k^*)})^2}{16}\right) \leqslant \int_{\tau > 16N_{k,t}^*} \exp\left(-\frac{\tau(\Delta_k^{(\tilde{m}_k^*)})^2}{16}\right) d\tau \leqslant \frac{16\delta}{(\Delta_k^{(\tilde{m}_k^*)})^2 Lt^4},$$

where inequality (a) is due to the definition of $c$, and inequality (b), denoting $\hat{\mu}_{k,t(\tau)}^{(m_k^*)}$ as the empirical mean of $\tau$ observations, is due to $\beta(\tau, t) < \frac{\Delta_k}{4}$ for $\tau > 16N_{k,t}^*$.

From Step 3, we obtain that the following equation holds with high probability,

$$N_{k,t}^{(\tilde{m}_k^*)} \leqslant \frac{16}{(\Delta_k^{(\tilde{m}_k^*)})^2} \log\left(\frac{Lt^4}{\delta}\right) \leqslant \frac{64}{(\Delta_k^{(\tilde{m}_k^*)})^2} \log\left(\frac{Lt}{\delta}\right). \tag{14}$$

Next, we respectively present the cost complexity upper bounds for different fidelity selection procedures in Algorithm 2.

### E.2.1 Proof for EXPLORE-A's Upper Bound

**Step 4 for EXPLORE-A: prove that if the small probability events of Steps 2 and 3 do not happen, then the algorithm terminates with a high probability when $\Lambda$ is large.**

**Lemma E.2.** *Give reward means $\mu_1^{(M)}$ and $\mu_2^{(M)}$. For a fixed arm $k$, there exist $\bar{N}_{k,t}$ and $\alpha_k > 0$ such that when $N_{k,t} > \bar{N}_{k,t}$, $N_{k,t} < 2N_{k,t}^{(\tilde{m}_k^*)}$, the number of times of pulling this arm $k$ at fidelities $m\,(\neq \tilde{m}_k^*)$ is $O(\log(\log N_{k,t}))$, or formally,*

$$N_{k,t}^{(m)} \leqslant \frac{8}{\lambda^{(m)}} \left( \frac{\Delta_k^{(\tilde{m}_k^*)}}{\sqrt{\lambda^{(\tilde{m}_k^*)}}} - \frac{\Delta_k^{(m)}}{\sqrt{\lambda^{(m)}}} \right)^{-2} \log N_{k,t}, \; \forall m \neq \tilde{m}_k^*. \tag{15}$$

Combine Lemma E.2 with Eq.(14) in Step 3, we have, for any arm $k$ and fidelity $m \neq \tilde{m}_k^*$:

$$N_{k,t}^{(m)} \leqslant \frac{8}{\lambda^{(m)}} \left( \frac{\Delta_k^{(\tilde{m}_k^*)}}{\sqrt{\lambda^{(\tilde{m}_k^*)}}} - \frac{\Delta_k^{(m)}}{\sqrt{\lambda^{(m)}}} \right)^{-2} \log \left( \frac{128}{(\Delta_k^{(\tilde{m}_k^*)})^2} \log \left( \frac{Lt}{\delta} \right) \right) \tag{16}$$

Next, we can upper bound the total cost of the LUCB algorithm (before it terminating) via Eq.(14) and Eq.(16). Specially, we show it is impossible for $\Lambda = C \left( H \log \frac{L(G+H)}{\lambda^{(1)}\delta} + G \log \log \frac{L(G+H)}{\lambda^{(1)}\delta} \right)$ via contradiction. Suppose $\Lambda = C \left( H \log \frac{L(G+H)}{\lambda^{(1)}\delta} + G \log \log \frac{L(G+H)}{\lambda^{(1)}\delta} \right)$, we have the following,

$$
\begin{aligned}
\mathbb{E}[\Lambda] &\leqslant \sum_{k \in \mathcal{K}} \sum_{m \in \mathcal{M}} \lambda^{(m)} N_{k,t}^{(m)} \\
&\leqslant \sum_{k \in \mathcal{K}} \lambda^{(\tilde{m}_k^*)} N_{k,t}^{(\tilde{m}_k^*)} + \sum_{k \in \mathcal{K}} \sum_{m \neq \tilde{m}_k^*} \lambda^{(m)} N_{k,t}^{(m)} \\
&\overset{(a)}{\leqslant} 64H \log \frac{Lt}{\delta} + 8G \log \log \frac{Lt}{\delta} + G \log(128H) \\
&\overset{(b)}{\leqslant} 64H \log \frac{L\Lambda}{\lambda^{(1)}\delta} + 8G \log \log \frac{L\Lambda}{\lambda^{(1)}\delta} + G \log(128H) \\
&\overset{(c)}{=} 64H \log \left( \frac{L}{\lambda^{(1)}\delta} C \left( H \log \frac{L(G+H)}{\lambda^{(1)}\delta} + G \log \log \frac{L(G+H)}{\lambda^{(1)}\delta} \right) \right) \\
&\quad + 8G \log \log \left( \frac{L}{\lambda^{(1)}\delta} C \left( H \log \frac{L(G+H)}{\lambda^{(1)}\delta} + G \log \log \frac{L(G+H)}{\lambda^{(1)}\delta} \right) \right) + G \log(128H) \\
&\overset{(d)}{\leqslant} 128(2 + \log C) \left( H \log \frac{L(G+H)}{\lambda^{(1)}\delta} + G \log \log \frac{L(G+H)}{\lambda^{(1)}\delta} \right) \\
&\overset{(e)}{<} C \left( H \log \frac{L(G+H)}{\lambda^{(1)}\delta} + G \log \log \frac{L(G+H)}{\lambda^{(1)}\delta} \right),
\end{aligned}
$$

where the inequality (a) is due to Eq.(14) and Eq.(16), the inequality (b) is because $t \leqslant \frac{\Lambda}{\lambda^{(1)}}$, the inequality (c) is by the supposition, the inequality (d) is by separately bounding the above first two terms via Eq.(17) and Eq.(18) in the following, and the inequality (e) holds for $C > 1200$. This above inequality contradicts the supposition, and, therefore, we conclude the cost complexity upper bound proof for EXPLORE-A. Similar proof also holds for EXPLORE-B by replacing $\tilde{m}_k^*$ with $m_k^\dagger$.

Next, we provide the upper bounds used in the inequality (d) above:

$$H \log \left( \frac{L}{\lambda^{(1)}\delta} C \left( H \log \frac{L(G+H)}{\lambda^{(1)}\delta} + G \log \log \frac{L(G+H)}{\lambda^{(1)}\delta} \right) \right)$$

$$\leqslant H \log \left( \frac{L}{\lambda^{(1)}\delta} CH \log \frac{L(G+H)}{\lambda^{(1)}\delta} \right) + H \log \left( \frac{L}{\lambda^{(1)}\delta} CG \log \log \frac{L(G+H)}{\lambda^{(1)}\delta} \right)$$

$$\leqslant H \log C + H \log \left( \frac{LH}{\lambda^{(1)}\delta} \log \frac{L(G+H)}{\lambda^{(1)}\delta} \right) + H \log C + H \log \left( \frac{LG}{\lambda^{(1)}\delta} \log \log \frac{L(G+H)}{\lambda^{(1)}\delta} \right)$$

$$\leqslant 2H \log C + 4H \log \frac{L(G+H)}{\lambda^{(1)}\delta}$$

$$\leqslant (4 + 2\log C) H \log \frac{L(G+H)}{\lambda^{(1)}\delta}, \tag{17}$$

and

$$G \log \log \left( \frac{L}{\lambda^{(1)}\delta} C \left( H \log \frac{L(G+H)}{\lambda^{(1)}\delta} + G \log \log \frac{L(G+H)}{\lambda^{(1)}\delta} \right) \right)$$

$$\leqslant G \log \log \left( \frac{L}{\lambda^{(1)}\delta} CH \log \frac{L(G+H)}{\lambda^{(1)}\delta} \right) + G \log \log \left( \frac{L}{\lambda^{(1)}\delta} CG \log \log \frac{L(G+H)}{\lambda^{(1)}\delta} \right)$$

$$\leqslant G \log \log C + G \log \log \left( \frac{LH}{\lambda^{(1)}\delta} \log \frac{L(G+H)}{\lambda^{(1)}\delta} \right) \tag{18}$$

$$\quad + G \log \log C + G \log \log \left( \frac{LG}{\lambda^{(1)}\delta} \log \log \frac{L(G+H)}{\lambda^{(1)}\delta} \right)$$

$$\leqslant 2G \log \log C + 4G \log \log \frac{L(G+H)}{\lambda^{(1)}\delta}$$

$$\leqslant (4 + 2\log \log C) G \log \log \frac{L(G+H)}{\lambda^{(1)}\delta}.$$

*Proof of Lemma E.2.*

**Claim 1.** For any fixed $m \neq \tilde{m}_k^*$, if the following equation holds, then the algorithm will not pull arm $k$ at fidelity $m$ with high probability.

$$N_{k,t}^{(m)} > \frac{8}{\lambda^{(m)}} \left( \frac{\Delta_k^{(\tilde{m}_k^*)}}{\sqrt{\lambda^{(\tilde{m}_k^*)}}} - \frac{\Delta_k^{(m)}}{\sqrt{\lambda^{(m)}}} \right)^{-2} \log N_{k,t}.$$

$$\texttt{f-UCB}_{u_t,t}^{(m)}(\mu_1^{(M)}) = \frac{1}{\sqrt{\lambda^{(m)}}} \left( \mu_1^{(M)} - \hat{\mu}_{u_t,t}^{(m)} - \zeta^{(m)} + \sqrt{\frac{2\log N_{u_t,t}}{N_{u_t,t}^{(m)}}} \right)$$

$$\overset{(a)}{\leqslant} \frac{1}{\sqrt{\lambda^{(m)}}} \left( \mu_1^{(M)} - \mu_{u_t}^{(m)} - \zeta^{(m)} + 2\sqrt{\frac{2\log N_{u_t,t}}{N_{u_t,t}^{(m)}}} \right)$$

$$\overset{(b)}{\leqslant} \frac{1}{\sqrt{\lambda^{(m_{u_t}^*)}}} \left( \mu_1^{(M)} - \mu_{u_t}^{(m_{u_t}^*)} - \zeta^{(m_{u_t}^*)} \right)$$

$$\overset{(c)}{\leqslant} \frac{1}{\sqrt{\lambda^{(\tilde{m}_k^*)}}} \left( \mu_1^{(M)} - \hat{\mu}_{u_t,t}^{(m_{u_t}^*)} - \zeta^{(m_{u_t}^*)} + \sqrt{\frac{2\log N_{u_t,t}}{N_{u_t,t}^{(\tilde{m}_k^*)}}} \right)$$

$$= \texttt{f-UCB}_{u_t,t}^{(m_{u_t}^*)}(\mu_1^{(M)}),$$

where the inequalities (a) and (c) hold with a probability at least $1 - \frac{1}{(N_{k,t})^2}$ respectively (by Hoeffding's inequality), and the inequality (b) holds due to the equation in the claim.

### E.2.2 Proof for EXPLORE-B's Upper Bound

As EXPLORE-B of Algorithm 2 also employs the LUCB framework, it shares the first three steps of the proof for Theorem 3.4 in Appendix E.2. Hence, in this part, we focus on the proof of the final cost complexity upper bound.

**Lemma E.3.** *If the condition in Line 10 holds, then the committed fidelity $\hat{m}_k^*$ fulfills the following inequality:*

$$2 \cdot \frac{\Delta_k^{(\hat{m}_k^*)}}{\sqrt{\lambda^{(\hat{m}_k^*)}}} \geqslant \frac{\Delta_k^{(\tilde{m}_k^*)}}{\sqrt{\lambda^{(\tilde{m}_k^*)}}}. \tag{19}$$

*Proof of Lemma E.3.* Eq.(19) is proved as follows,

$$
\begin{aligned}
\frac{\Delta_k^{(\tilde{m}_k^*)}/\sqrt{\lambda^{(\tilde{m}_k^*)}}}{\Delta_k^{(\hat{m}_k^*)}/\sqrt{\lambda^{(\hat{m}_k^*)}}} &\leqslant \frac{(\hat{\mu}_*^{(M)} - (\hat{\mu}_k^{(\tilde{m}_k^*)} + \zeta^{(\tilde{m}_k^*)}))/\sqrt{\lambda^{(\tilde{m}_k^*)}} + \sqrt{\log(2KM/\delta)/\lambda^{(1)} N_{k,t}^{(m)}}}{(\hat{\mu}_*^{(M)} - (\hat{\mu}_k^{(\hat{m}_k^*)} + \zeta^{(\hat{m}_k^*)}))/\sqrt{\lambda^{(\hat{m}_k^*)}} - \sqrt{\log(2KM/\delta)/\lambda^{(1)} N_{k,t}^{(m)}}} \\[2mm]
&\overset{(a)}{\leqslant} \frac{(\hat{\mu}_*^{(M)} - (\hat{\mu}_k^{(\hat{m}_k^*)} + \zeta^{(\hat{m}_k^*)}))/\sqrt{\lambda^{(\hat{m}_k^*)}} + \sqrt{\log(2KM/\delta)/\lambda^{(1)} N_{k,t}^{(m)}}}{(\hat{\mu}_*^{(M)} - (\hat{\mu}_k^{(\hat{m}_k^*)} + \zeta^{(\hat{m}_k^*)}))/\sqrt{\lambda^{(\hat{m}_k^*)}} - \sqrt{\log(2KM/\delta)/\lambda^{(1)} N_{k,t}^{(m)}}} \\[2mm]
&\leqslant 1 + \frac{2\sqrt{\log(2KM/\delta)/\lambda^{(1)} N_{k,t}^{(m)}}}{(\hat{\mu}_*^{(M)} - (\hat{\mu}_k^{(\hat{m}_k^*)} + \zeta^{(\hat{m}_k^*)}))/\sqrt{\lambda^{(\hat{m}_k^*)}} - \sqrt{\log(2KM/\delta)/\lambda^{(1)} N_{k,t}^{(m)}}} \\[2mm]
&\overset{(b)}{\leqslant} 1 + \frac{2\sqrt{\log(2KM/\delta)/\lambda^{(1)} N_{k,t}^{(m)}}}{2\sqrt{\log(2KM/\delta)/\lambda^{(1)} N_{k,t}^{(m)}}} = 2,
\end{aligned}
$$

where inequality (a) is due to the definition of $\hat{m}_k^*$, and inequality (b) is due to the condition in Line 10. $\qquad\square$

We next upper bound the number of times of $N_{k,t}^{(m)}$ that guarantees that the condition in Line 10 is true. Let us consider the case of exploring arm $u_t$.

$$
\begin{aligned}
\max_{m \in \mathcal{M}} \frac{\hat{\Delta}_{k,t}^{(m)}}{\sqrt{\lambda^{(m)}}} &\geqslant \frac{\hat{\mu}_{k_*}^{(M)} - (\hat{\mu}_k^{(\tilde{m}_k^*)} + \zeta^{(\tilde{m}_k^*)})}{\sqrt{\lambda^{(\tilde{m}_k^*)}}} \\[2mm]
&\overset{(a)}{\geqslant} \frac{\hat{\mu}_{k_*}^{(M)} - (\mu_k^{(\tilde{m}_k^*)} + \zeta^{(\tilde{m}_k^*)})}{\sqrt{\lambda^{(\tilde{m}_k^*)}}} - \sqrt{\frac{\log(2KM/\delta)}{\lambda^{(1)} N_{k,t}^{(m)}}} \\[2mm]
&\overset{(b)}{\geqslant} \frac{\Delta_k^{(\tilde{m}_k^*)}}{\sqrt{\lambda^{(\tilde{m}_k^*)}}} - \sqrt{\frac{\log(2KM/\delta)}{\lambda^{(1)} N_{k,t}^{(m)}}},
\end{aligned}
$$

where inequality (a) is because that $\hat{\mu}_k^{(\tilde{m}_k^*)} \leqslant \mu_k^{(\tilde{m}_k^*)} + \sqrt{\frac{\log(2KM/\delta)}{N_{k,t}^{(m)}}}$ with a probability of at least $1 - \delta/2KM$ (therefore, with the union bound over all arm-fidelity pairs, the total failure probability of EXPLORE is upper bounded by $\delta/2$), and inequality (b) is because $\hat{\mu}_{k_*}^{(M)} - (\mu_k^{(\tilde{m}_k^*)} + \zeta^{(\tilde{m}_k^*)}) \geqslant \mu_{k_*}^{(M)} - (\mu_k^{(\tilde{m}_k^*)} + \zeta^{(\tilde{m}_k^*)}) = \Delta_k^{(\tilde{m}_k^*)}$.

To make the condition in Line 10 hold, with the above inequality, we need

$$\frac{\Delta_k^{(\tilde{m}_k^*)}}{\sqrt{\lambda^{(\tilde{m}_k^*)}}} - \sqrt{\frac{\log(2KM/\delta)}{\lambda^{(1)} N_{k,t}^{(m)}}} \geqslant 3\sqrt{\frac{\log(2KM/\delta)}{\lambda^{(1)} N_{k,t}^{(m)}}},$$

which, after rearrangement, becomes

$$N_{k,t}^{(m)} > \frac{16\lambda^{(\tilde{m}_k^*)}}{(\Delta_k^{(\tilde{m}_k^*)})^2} \frac{\log(2KM/\delta)}{\lambda^{(1)}}.$$

It means that if the above inequality holds, than the condition in Line 10 must hold. That is, except for the committed fidelity $\hat{m}_k^*$, we have

$$N_{k,t}^{(m)} \leqslant \frac{16\lambda^{(\tilde{m}_k^*)}}{(\Delta_k^{(\tilde{m}_k^*)})^2} \frac{\log(2KM/\delta)}{\lambda^{(1)}}, \text{ for any other fidelities } m \neq \hat{m}_k^*.$$

For another thing, Eq.(14) of LUCB's proof guarantees that for the selected fidelity $\hat{m}_k^*$, the number of pulling times is upper bounded as follows,

$$N_{k,t}^{(\hat{m}_k^*)} \leqslant \frac{64}{(\Delta_k^{(\hat{m}_k^*)})^2} \log\left(\frac{Lt}{\delta}\right).$$

Then, we upper bound the total budget of the algorithm as follows,

$$
\begin{aligned}
\Lambda &= \sum_{k\in\mathcal{K}} \sum_{m\in\mathcal{M}} \lambda^{(m)} N_{k,t}^{(m)} \\
&\leqslant \sum_{k\in\mathcal{K}} \frac{64\lambda^{(\hat{m}_k^*)}}{(\Delta_k^{(\hat{m}_k^*)})^2} \log\left(\frac{Lt}{\delta}\right) + \sum_{k\in\mathcal{K}} \sum_{m\neq\hat{m}_k^*} \frac{16\lambda^{(m)}\lambda^{(\tilde{m}_k^*)}}{(\Delta_k^{(\tilde{m}_k^*)})^2 \lambda^{(1)}} \log\left(\frac{2KM}{\delta}\right) \\
&\overset{(a)}{\leqslant} \sum_{k\in\mathcal{K}} \frac{256\lambda^{(\tilde{m}_k^*)}}{(\Delta_k^{(\tilde{m}_k^*)})^2} \log\left(\frac{Lt}{\delta}\right) + \sum_{k\in\mathcal{K}} \sum_{m\neq\hat{m}_k^*} \frac{16\lambda^{(m)}\lambda^{(\tilde{m}_k^*)}}{(\Delta_k^{(\tilde{m}_k^*)})^2 \lambda^{(1)}} \log\left(\frac{2KM}{\delta}\right) \\
&\leqslant \left( \sum_{k\in\mathcal{K}} \frac{256\lambda^{(\tilde{m}_k^*)}}{(\Delta_k^{(\tilde{m}_k^*)})^2} + \sum_{k\in\mathcal{K}} \sum_{m\neq\hat{m}_k^*} \frac{16\lambda^{(m)}\lambda^{(\tilde{m}_k^*)}}{(\Delta_k^{(\tilde{m}_k^*)})^2 \lambda^{(1)}} \right) \log\left(\frac{Lt}{\delta}\right) \\
&\leqslant \sum_{m\in\mathcal{M}} \frac{\lambda^{(m)}}{\lambda^{(1)}} \cdot \sum_{k\in\mathcal{K}} \frac{256\lambda^{(\tilde{m}_k^*)}}{(\Delta_k^{(\tilde{m}_k^*)})^2} \log\left(\frac{Lt}{\delta}\right) \\
&\leqslant \sum_{m\in\mathcal{M}} \frac{\lambda^{(m)}}{\lambda^{(1)}} \cdot \sum_{k\in\mathcal{K}} \frac{256\lambda^{(\tilde{m}_k^*)}}{(\Delta_k^{(\tilde{m}_k^*)})^2} \log\left(\frac{L\Lambda}{\delta\lambda^{(1)}}\right) \\
&\overset{(b)}{\leqslant} \sum_{m\in\mathcal{M}} \frac{\lambda^{(m)}}{\lambda^{(1)}} \cdot \sum_{k\in\mathcal{K}} \frac{1024\lambda^{(\tilde{m}_k^*)}}{(\Delta_k^{(\tilde{m}_k^*)})^2} \log\left( \sum_{m\in\mathcal{M}} \frac{\lambda^{(m)}}{\lambda^{(1)}} \cdot \sum_{k\in\mathcal{K}} \frac{256\lambda^{(\tilde{m}_k^*)}}{(\Delta_k^{(\tilde{m}_k^*)})^2} \frac{L}{\lambda^{(1)}\delta} \right) \\
&\leqslant O\left( \sum_{m\in\mathcal{M}} \frac{\lambda^{(m)}}{\lambda^{(1)}} \cdot \tilde{H} \log\left( \sum_{m\in\mathcal{M}} \frac{\lambda^{(m)}}{\lambda^{(1)}} \cdot \tilde{H} \cdot \frac{L}{\lambda^{(1)}\delta} \right) \right),
\end{aligned}
$$

where inequality (a) is due to Eq.(19), inequality (b) is due to that $\Lambda \leqslant A\log(B\Lambda) \implies \Lambda \leqslant 4A\log(AB\Lambda)$.

### E.2.3   Proof for EXPLORE-C's Upper Bound in Theorem D.1

**Step 4 for EXPLORE-C: Prove that if the events of Steps 2 and 3 do not happen, for $\Lambda > O\left( Q\log\left(\frac{KM\sqrt{Q}}{\delta(\lambda^{(1)})^2}\right) \right)$, the algorithm terminates with a probability at least $O(1-\delta/\Lambda^2)$.** Denote $\bar{T} := \lceil \frac{\Lambda}{2\lambda^{(M)}} \rceil$ and two events $E_1, E_2$ as follows,

$$
\begin{aligned}
E_1 &:= \{\exists t \geqslant \bar{T} : \mathtt{CROS}_t\}, \\
E_2 &:= \{\exists t \geqslant \bar{T}, k \in \mathcal{K} : (n_{k,t}^{(m_k^*)} > 16n_{k,t}^*) \cup (k \in \mathtt{Mid}_t)\}.
\end{aligned}
$$

We first upper bound the number of rounds after $\bar{T}$ as follows,

$$
\begin{aligned}
\sum_{t \geqslant \bar{T}} \lambda^{(m_t)} \mathbb{1}\{\neg \texttt{TERM}_t\} & \overset{(a)}{=} \sum_{t \geqslant \bar{T}} \lambda^{(m_t)} \mathbb{1}\{\neg \texttt{TERM}_t \cap \neg \texttt{CROS}_t\} \\
& \overset{(b)}{\leqslant} \sum_{t \geqslant \bar{T}} \lambda^{(m_t)} \mathbb{1}\{(\ell_t \in \texttt{Mid}_t) \cup (u_t \in \texttt{Mid}_t)\} \\
& \leqslant \sum_{t \geqslant \bar{T}} \sum_{k \in \mathcal{K}} \lambda^{(m_t)} \mathbb{1}\{((k = \ell_t) \cup (k = u_t)) \cap (k \in \texttt{Mid}_t)\} \\
& \overset{(c)}{\leqslant} \sum_{t \geqslant \bar{T}} \sum_{k \in \mathcal{K}} \lambda^{(m_t)} \mathbb{1}\Big\{((k = \ell_t) \cup (k = u_t)) \cap (n_{k,t}^{(m_k^*)} \leqslant 16 n_{k,t}^*)\Big\} \\
& = \sum_{k \in \mathcal{K}} \sum_{t \geqslant \bar{T}} \lambda^{(m_t)} \mathbb{1}\Big\{((k = \ell_t) \cup (k = u_t)) \cap (n_{k,t}^{(m_k^*)} \leqslant 16 n_{k,t}^*)\Big\} \\
& \leqslant \sum_{k \in \mathcal{K}} \left( \sum_{\ell=1}^{m_k^*-1} \frac{\lambda^{(\ell)} \log(Lt^4/\delta)}{(\zeta^{(\ell)})^2} + 16 \lambda^{(m_k^*)} n_{k,t}^* \right) \\
& \leqslant \sum_{k \in \mathcal{K}} \left( \sum_{\ell=1}^{m_k^*-1} \frac{\lambda^{(\ell)}}{(\zeta^{(\ell)})^2} + \frac{16 \lambda^{(m_k^*)}}{\Delta_k^2} \right) \log \left( \frac{LT^4}{\delta} \right) \\
& \leqslant 4 \sum_{k \in \mathcal{K}} \left( \sum_{\ell=1}^{m_k^*-1} \frac{\lambda^{(\ell)}}{(\zeta^{(\ell)})^2} + \frac{16 \lambda^{(m_k^*)}}{\Delta_k^2} \right) \log \left( \frac{LT}{\delta} \right) \\
& \leqslant 4 \sum_{k \in \mathcal{K}} \left( \sum_{\ell=1}^{m_k^*-1} \frac{\lambda^{(\ell)}}{(\zeta^{(\ell)})^2} + \frac{16 \lambda^{(m_k^*)}}{\Delta_k^2} \right) \log \left( \frac{L\Lambda}{\lambda^{(1)}\delta} \right)
\end{aligned}
\tag{20}
$$

where the equation (a) is due to $\neg E_1$, the inequality (b) is due to Step 1, and the inequality (c) is due to $\neg E_2$.

Also notice that

$$
\Lambda = \sum_{t < \bar{T}} \lambda^{(m_t)} + \sum_{t \geqslant \bar{T}} \lambda^{(m_t)} \mathbb{1}\{\neg \texttt{TERM}_t\} \leqslant \frac{\Lambda}{2} + \sum_{t \geqslant \bar{T}} \lambda^{(m_t)} \mathbb{1}\{\neg \texttt{TERM}_t\},
$$

and, combining with Eq.(20), we have,

$$
\Lambda \leqslant 8 \sum_{k \in \mathcal{K}} \left( \sum_{\ell=1}^{m_k^*-1} \frac{\lambda^{(\ell)}}{(\zeta^{(\ell)})^2} + \frac{16 \lambda^{(m_k^*)}}{\Delta_k^2} \right) \log \left( \frac{L\Lambda}{\lambda^{(1)}\delta} \right).
$$

Solving the above inequality concludes the cost complexity upper bound for EXPLORE-C.

In the end of Step 4, we show that the LUCB algorithm fulfills the fixed confidence requirement. The probability that the algorithm does not terminate after spending $\Lambda$ budget is upper bounded by $\mathbb{P}(E_1 \cup E_2)$. Based on Steps 2 and 3, it can be upper bounded as follows,

$$
\begin{aligned}
\mathbb{P}(E_1 \cup E_2) & \leqslant \sum_{t > \bar{T}} \left( \frac{KM\delta}{Lt^3} + \frac{16\delta \sum_{k \in \mathcal{K}} \Delta_k^{-2}}{Lt^4} \right) \leqslant \Lambda \left( \frac{1}{\lambda^{(1)}} - \frac{1}{2\lambda^{(M)}} \right) \left( \frac{\delta}{(\Lambda/\lambda^{(1)})^3} \right) \sum_{k \in \mathcal{K}} \Delta_k^{-2} \\
& \leqslant \frac{\delta}{\Lambda^2} \sum_{k \in \mathcal{K}} \Delta_k^{-2} \left( (\lambda^{(1)})^2 - \frac{(\lambda^{(1)})^3}{2\lambda^{(M)}} \right) \\
& \leqslant \delta.
\end{aligned}
$$

$\square$

# F Proofs for Regret Minimization Results

The two lower bound proofs utilize two different regret decomposition as follows,

$$R(\Lambda) \overset{(a)}{\geqslant} \sum_{t=1}^{N} \left( \frac{\lambda^{(m_t)} - \lambda^{(1)}}{\lambda^{(1)}} \mu_1 + \Delta_{I_t}^{(M)} \right)$$

$$= \sum_{m=2}^{M} N_{\forall k}^{(m)}(\Lambda) \frac{\lambda^{(m)} - \lambda^{(1)}}{\lambda^{(1)}} \mu_1 + \sum_{k \neq k_1^{(M)}} N_k^{(\forall m)}(\Lambda) \Delta_k^{(M)}; \tag{21}$$

$$R(\Lambda) = \sum_{k \in \mathcal{K}} \sum_{m=1}^{M} N_k^{(m)}(\Lambda) \left( \frac{\lambda^{(m)}}{\lambda^{(1)}} \mu_1^{(M)} - \mu_k^{(M)} \right), \tag{22}$$

where the inequality (a) is due to $\Lambda > \sum_{t=1}^{N} \lambda^{(m_t)}$, the $N_k^{(m)}(\Lambda)$ is the number of times that arm $k$ is pulled in fidelity $m$ after paying budget $\Lambda$. The $N$ with subscript $_{\forall k}$ and superscript $^{(\forall m)}$ mean the pulling times of all arms and all fidelities respectively. Due to the multi-fidelity feedback, both decompositions are different from classic bandits' regret decomposition [24, Lemma 4.5], and, therefore, we need to non-trivially extend the known approaches to our scenario.

To prove the problem-independent lower bound $\Omega(K^{1/3}\Lambda^{2/3})$ in Theorem 4.3, we utilize the decomposition in Eq.(21): In the RHS, the first term increases whether one choose fidelity other than the lowest; this is a novel term. The second term of RHS corresponds to the cost of pulling suboptimal arms which also appears in the classic MAB. With this observation, one can construct instance pairs such that it is unavoidable to do exploration at higher fidelities and, therefore, the first term is not negligible. Lastly, one needs to balance the magnitude of the above two terms case-by-case, which together bounds the regret as $\Omega\left(\Lambda^{2/3}\right)$. To prove the problem-dependent lower bound in Theorem F.1, we utilize Eq.(22) to decompose the regret to each arm and bound each of them separately.

## F.1 Proof of Theorem 4.3

**Step 1. Construct instances and upper bound KL-divergence**  Fix a policy $\pi$. We construct two MF-MAB instances, each with $K$ arms and $M$ fidelities. For the pulling costs of different fidelities, we set $\lambda^{(M)} \leqslant 2\lambda^{(1)}$. For reward feedback, we assume all arms' reward distributions at any fidelity are Bernoulli, and denote $\mu_k^{(m)}(1), \mu_k^{(m)}(2)$ as the reward means of these two instances. Let $\mathbb{P}_1 = \mathbb{P}_{\boldsymbol{\mu}(1),\pi}, \mathbb{E}_1 = \mathbb{E}_{\boldsymbol{\mu}(1),\pi}$ and $\mathbb{P}_2 = \mathbb{P}_{\boldsymbol{\mu}(2),\pi}, \mathbb{E}_2 = \mathbb{E}_{\boldsymbol{\mu}(2),\pi}$ be the probability measures and expectations on the canonical MF-MAB model induced by $\Lambda$-budget interconnection of $\pi$ and $\boldsymbol{\mu}_1$ (and $\boldsymbol{\mu}_2$). Denote $k' = \arg\min_{k \in \mathcal{K}} \mathbb{E}_1[N_k^{(m>1)}(\Lambda)]$.

The only difference between both instance pair is in the arm $k'$'s reward mean for fidelities $m > 1$, so that instance 1's optimal arm is arm 1 and instance 2's optimal arm is arm $k'$. The detailed reward means are listed as follows,

| | $m = 1$ | $m > 1$ |
|---|---|---|
| $(\mu_k^{(m)}(1))_{k \in \mathcal{K}}$ | $\left(\frac{1}{2} + \Delta, \frac{1}{2}, \frac{1}{2}, \ldots, \frac{1}{2}\right)$ | $\left(\frac{1}{2} + \Delta, \frac{1}{2}, \frac{1}{2}, \ldots, \frac{1}{2}\right)$ |
| $(\mu_k^{(m)}(2))_{k \in \mathcal{K}}$ | $\left(\frac{1}{2} + \Delta, \frac{1}{2}, \ldots, \frac{1}{2}\right)$ | $\left(\frac{1}{2} + \Delta, \frac{1}{2}, \ldots, \frac{1}{2}, \frac{1}{2} + 2\Delta, \frac{1}{2}, \ldots, \frac{1}{2}\right)$ |

Denote the entry $(M_t, I_t, X_t)$ as a tuple of the pulled fidelity, pulled arm, and observed reward random variables at time $t$, and $\mathcal{H} := (M_1, I_1, X_1; M_2, I_2, X_2; \ldots; M_N, I_N, X_N)$ as a sequence of applying policy $\pi$. We note that $N$ is also a random variable depending on the sequence of fidelities in pulling arms. Next, we calculate the upper bound of the KL-divergence of the above two instances in this sequence.

$$\mathrm{KL}(\mathbb{P}_1, \mathbb{P}_2) = \mathbb{E}_1 \left[ \log \left( \frac{d\mathbb{P}_1}{d\mathbb{P}_2} \right) \right]$$

$$= \mathbb{E}_1 \left[ \log \left( \frac{d\mathbb{P}_1}{d\mathbb{P}_2} (M_1, I_1, X_1; M_2, I_2, X_2; \ldots; M_N, I_N, X_N) \right) \right]$$

$$= \mathbb{E}_1 \left[ \mathbb{E}_1 \left[ \log \left( \frac{d\mathbb{P}_1}{d\mathbb{P}_2} (M_1, I_1, X_1; M_2, I_2, X_2; \ldots; M_N, I_N, X_N) \right) \Big| N \right] \right]$$

$$\stackrel{(a)}{=} \mathbb{E}_1 \left[ \mathbb{E}_1 \left[ \sum_{t=1}^{N} \log \left( \frac{p_1(X_t|M_t, I_t)}{p_2(X_t|M_t, I_t)} \right) \Big| N \right] \right]$$

$$= \mathbb{E}_1 \left[ \sum_{t=1}^{N} \mathbb{E}_1 \left[ \log \left( \frac{p_1(X_t|M_t, I_t)}{p_2(X_t|M_t, I_t)} \right) \Big| N \right] \right]$$

$$\stackrel{(b)}{=} \mathbb{E}_1 \left[ \sum_{t=1}^{N} \mathbb{E}_1 \left[ \mathrm{KL} \left( P_1^{(M_t, I_t)}, P_2^{(M_t, I_t)} \right) \Big| N \right] \right]$$

$$= \sum_{(m,k) \in \mathcal{M} \times \mathcal{K}} \mathbb{E}_1 \left[ \sum_{t=1}^{N} \mathbb{E}_1 \left[ \mathbb{1}\{M_t = m, I_t = k\} \, \mathrm{KL} \left( P_1^{(M_t, I_t)}, P_2^{(M_t, I_t)} \right) \Big| N \right] \right]$$

$$= \sum_{(m,k) \in \mathcal{M} \times \mathcal{K}} \mathbb{E}_1 \left[ \mathrm{KL} \left( P_1^{(m,k)}, P_2^{(m,k)} \right) \sum_{t=1}^{N} \mathbb{E}_1 \left[ \mathbb{1}\{M_t = m, I_t = k\} | N \right] \right]$$

$$= \sum_{(m,k) \in \mathcal{M} \times \mathcal{K}} \mathrm{KL} \left( P_1^{(m,k)}, P_2^{(m,k)} \right) \mathbb{E}_1 \left[ \mathbb{E}_1 \left[ \sum_{t=1}^{N} \mathbb{1}\{M_t = m, I_t = k\} \Big| N \right] \right]$$

$$\stackrel{(c)}{=} \sum_{(m,k) \in \mathcal{M} \times \mathcal{K}} \mathrm{KL} \left( P_1^{(m,k)}, P_2^{(m,k)} \right) \mathbb{E}_1 \left[ T_k^{(m)}(\Lambda) \right]$$

$$= \mathrm{KL} \left( P_1^{(m,2)}, P_2^{(m,2)} \right) \mathbb{E}_1 \left[ T_{k'}^{(m>1)}(\Lambda) \right]$$

$$\stackrel{(d)}{\leqslant} \frac{1}{K} \cdot \mathbb{E}_1 \left[ T_{\forall k}^{(m>1)}(\Lambda) \right] \cdot \mathrm{KL} \left( \mathcal{B} \left( \frac{1}{2} \right), \mathcal{B} \left( \frac{1}{2} + 2\Delta \right) \right)$$

$$\stackrel{(e)}{\leqslant} \mathbb{E}_1 \left[ T_{\forall k}^{(m>1)}(\Lambda) \right] \cdot \frac{9\Delta^2}{K},$$

$$(23)$$

where the equation (a) is due to

$$\frac{d\mathbb{P}_1}{d(\rho \times \rho \times \lambda)^N} (m_1, k_1, x_1; m_2, k_2, x_2; \ldots; m_N, k_N, x_N)$$

$$= p_{\boldsymbol{\mu}_1, \pi}(m_1, k_1, x_1; m_2, k_2, x_2; \ldots; m_N, k_N, x_N)$$

$$= \prod_{t=1}^{N} \pi_t(m_t, k_t | m_1, k_1, x_1; \ldots; m_{t-1}, k_{t-1}, x_{t-1}) p_{\boldsymbol{\mu}_1}(x_t | m_t, k_t),$$

the equation (b) is due to

$$\mathbb{E}_1 \left[ \log \left( \frac{p_1(X_t|M_t, I_t)}{p_2(X_t|M_t, I_t)} \right) \Big| N \right] = \mathbb{E}_1 \left[ \mathbb{E}_1 \left[ \log \left( \frac{p_1(X_t|M_t, I_t)}{p_2(X_t|M_t, I_t)} \right) \Big| M_t, I_t \right] \Big| N \right]$$

$$= \mathbb{E}_1 \left[ \mathrm{KL} \left( P_1^{(M_t, I_t)}, P_2^{(M_t, I_t)} \right) \Big| N \right],$$

the equation (c) is due to the tower property as well, the inequality (d) is because arm $k'$ is the arm with the smallest number of pulled with fidelity $m > 1$, and the inequality (e) is by calculating the KL-divergent between two Bernoulli distributions.

**Step 2. Lower bound the regret**  We first note that the regret defined in Eq.(9) for instance 1 can be decomposed as follows,

$$\mathbb{E}_1[R(\Lambda)] \overset{(a)}{\geqslant} \sum_{t=1}^{N} \left( \frac{\lambda^{(m_t)} - \lambda^{(1)}}{\lambda^{(1)}} \mu_1 + \Delta_{I_t}^{(M)} \right)$$

$$= \sum_{m=2}^{M} T_{\forall k}^{(m)}(\Lambda) \frac{\lambda^{(m)} - \lambda^{(1)}}{\lambda^{(1)}} \mu_1 + \sum_{k \neq k_1^{(M)}} T_k^{(\forall m)}(\Lambda) \Delta_k^{(M)}$$

where inequality (a) is due to $\Lambda > \sum_{t=1}^{N} \lambda^{(m_t)}$, the $T_k^{(m)}(\Lambda)$ is the number of times that arm $k$ is pulled in fidelity $m$ (given total budget $\Lambda$).

Then, we lower bound the summation of the regrets under both instances,

$$\mathbb{E}_1[R(\Lambda)] + \mathbb{E}_2[R(\Lambda)]$$

$$= \mathbb{E}_1 \left[ \sum_{m=2}^{M} T_{\forall k}^{(m)}(\Lambda) \frac{\lambda^{(m)} - \lambda^{(1)}}{\lambda^{(1)}} \mu_1 + \sum_{k \neq k_1^{(M)}} T_k^{(\forall m)}(\Lambda) \Delta_k^{(M)} \right]$$

$$+ \mathbb{E}_2 \left[ \sum_{m=2}^{M} T_{\forall k}^{(m)}(\Lambda) \frac{\lambda^{(m)} - \lambda^{(1)}}{\lambda^{(1)}} \mu_1 + \sum_{k \neq k_1^{(M)}} T_k^{(\forall m)}(\Lambda) \Delta_k^{(M)} \right]$$

$$\geqslant \mathbb{E}_1 \left[ T_{\forall k}^{(m>1)}(\Lambda) \frac{\lambda^{(2)} - \lambda^{(1)}}{\lambda^{(1)}} \mu_1 + \sum_{k \neq k_1^{(M)}} T_k^{(\forall m)}(\Lambda) \Delta_k^{(M)} \right]$$

$$+ \mathbb{E}_2 \left[ T_{\forall k}^{(m>1)}(\Lambda) \frac{\lambda^{(2)} - \lambda^{(1)}}{\lambda^{(1)}} \mu_1 + \sum_{k \neq k_1^{(M)}} T_k^{(\forall m)}(\Lambda) \Delta_k^{(M)} \right]$$

$$\overset{(a)}{\geqslant} \mathbb{E}_1 \left[ T_{\forall k}^{(m>1)}(\Lambda) \right] \frac{\lambda^{(2)} - \lambda^{(1)}}{\lambda^{(1)}} \mu_1$$

$$+ \Delta \min \left\{ \frac{\Lambda}{\lambda^{(M)}} - \frac{\Lambda}{2\lambda^{(1)}}, \frac{\Lambda}{2\lambda^{(1)}} \right\} \left( \mathbb{P}_1 \left( T_1^{(\forall m)} \leqslant \frac{\Lambda}{2\lambda^{(1)}} \right) + \mathbb{P}_2 \left( T_1^{(\forall m)} > \frac{\Lambda}{2\lambda^{(1)}} \right) \right)$$

$$\overset{(b)}{\geqslant} \mathbb{E}_1 \left[ T_{\forall k}^{(m>1)}(\Lambda) \right] \frac{\lambda^{(2)} - \lambda^{(1)}}{\lambda^{(1)}} \mu_1 + \frac{\Lambda \Delta}{2} \min \left\{ \frac{1}{\lambda^{(M)}} - \frac{1}{2\lambda^{(1)}}, \frac{1}{2\lambda^{(1)}} \right\} \exp(-\operatorname{KL}(\mathbb{P}_1, \mathbb{P}_2))$$

$$\overset{(c)}{\geqslant} \mathbb{E}_1 \left[ T_{\forall k}^{(m>1)}(\Lambda) \right] \frac{\lambda^{(2)} - \lambda^{(1)}}{\lambda^{(1)}} \mu_1$$

$$+ \frac{\Lambda \Delta}{2} \min \left\{ \frac{1}{\lambda^{(M)}} - \frac{1}{2\lambda^{(1)}}, \frac{1}{2\lambda^{(1)}} \right\} \exp \left( -2\Delta^2 K^{-1} \mathbb{E}_1 \left[ T_{\forall k}^{(m>1)}(\Lambda) \right] \right),$$

where inequality (a) uses the $\lambda^{(M)} \leqslant 2\lambda^{(1)}$ condition, inequality (b) is by Bretagnolle-Huber inequality [24, Theorem 14.2], and inequality (c) is by Eq.(23).

**Step 3. Obtain the $\Omega(K^{\frac{1}{3}} \Lambda^{\frac{2}{3}})$ lower bound**  We show that $\mathbb{E}_1[R(\Lambda)] + \mathbb{E}_2[R(\Lambda)] \geqslant \Omega(K^{\frac{1}{3}} \Lambda^{\frac{2}{3}})$ for any possible quantity of $\mathbb{E}_1 \left[ T_{\forall k}^{(m>1)}(\Lambda) \right]$ via categorized discussion as follows,

- Case 1: If $\mathbb{E}_1 \left[ T_{\forall k}^{(m>1)}(\Lambda) \right] = 0$, then the last formula becomes $C\Lambda\Delta$. Letting $\Delta$ be a constant (via sup), we have a $\Omega(\Lambda)$ lower bound, which means $\Omega(K^{\frac{1}{3}} \Lambda^{\frac{2}{3}})$ is also valid.

- Case 2: If $\mathbb{E}_1 \left[ T_{\forall k}^{(m>1)}(\Lambda) \right] \geqslant K^{\frac{1}{3}} \Lambda^{\frac{2}{3}}$, then we have the $\Omega(K^{\frac{1}{3}} \Lambda^{\frac{2}{3}})$ lower bound from the first term.

- Case 3: If $0 < \mathbb{E}_1 \left[ T_{\forall k}^{(m>1)}(\Lambda) \right] < K^{\frac{1}{3}} \Lambda^{\frac{2}{3}}$, we choose $\Delta = K^{\frac{1}{3}} \Lambda^{-\frac{1}{3}}$ and also obtain the $\Omega(K^{\frac{1}{3}} \Lambda^{\frac{2}{3}})$ lower bound.

### F.2 Proof of Theorem 4.4

In this proof, we prove that for any arm $k \in \mathcal{K}$, the total regret due to this arm $k$ is lower bounded as follows,

$$\liminf_{\Lambda \to \infty} \frac{\mathbb{E}\left[R_k(\Lambda)\right]}{\log(\Lambda)} \geqslant \min_{m:\Delta_k^{(m)}>0} \left(\frac{\lambda^{(m)}}{\lambda^{(1)}}\mu_1^{(M)} - \mu_k^{(M)}\right) \frac{C}{(\Delta_k^{(m)})^2}.$$

Then, noticing that $R(\Lambda) = \sum_{k \in \mathcal{M}} R_k(\Lambda)$ concludes the proof.

We construct instances $\nu$ and $\nu'$. We set the reward distributions $\nu = (\nu_k^{(m)})_{(k,m)\in\mathcal{K}\times\mathcal{M}}$ as Bernoulli and the reward means fulfill $\mu_1^{(M)} > \mu_2^{(M)} \geqslant \mu_3^{(M)} \geqslant \ldots \geqslant \mu_K^{(M)}$, where $\mu_k^{(m)} = \mathbb{E}_{X\sim\nu_k^{(m)}}[X]$. We let $\nu'_k^{(m)}$ be the same to $\nu_k^{(m)}$ for all $k$ and $m$, except for that an arm $\ell \neq 1$. We set arm $\ell$'s reward means on fidelities $m \in \mathcal{M}_k$ to be $\nu'_\ell^{(m)} = \nu_1^{(M)} - \zeta^{(m)} + \epsilon$. One can verify that the reward means of arm $k$ under instance $v'$ fulfill the condition that $|\mu'_k^{(m)} - \mu'_k^{(M)}| \leqslant \zeta^{(m)}$ for any fidelity $m$. Notice that $\mu'_k^{(M)} > \mu_{k_*}^{(M)}$. Hence, under instance $\mathcal{I}'$, the optimal arm is $k$. We denote $\mathbb{P}, \mathbb{E}$ and $\mathbb{P}', \mathbb{E}'$ as the probability measures and expectations for instances $\mathcal{I}$ and $\mathcal{I}'$ respectively.

Next, we employ the (extended) key inequality from Garivier et al. [12] as follows,

$$\sum_{k\in\mathcal{K}}\sum_{m\in\mathcal{M}} \mathbb{E}[N_k^{(m)}(\Lambda)]\,\mathrm{KL}(v_k^{(m)}, v'_k^{(m)}) \geqslant \mathrm{kl}(\mathbb{E}[Z], \mathbb{E}'[Z]).$$

Let $Z = \lambda^{(M)}N_\ell^{(\forall m)}(\Lambda)/\Lambda$ in the above inequality, we have

$$\sum_{m:\Delta_\ell^{(m)}>0} \mathbb{E}[N_k^{(m)}(\Lambda)]\,\mathrm{KL}(v_\ell^{(m)}, v'_\ell^{(m)})$$

$$\geqslant \mathrm{kl}\left(\frac{\mathbb{E}[N_\ell^{(\forall m)}(\Lambda)]}{\Lambda}, \frac{\mathbb{E}'[N_\ell^{(\forall m)}(\Lambda)]}{\Lambda}\right) \tag{24}$$

$$\overset{(a)}{\geqslant} \left(1 - \frac{\lambda^{(M)}\mathbb{E}[N_\ell^{(\forall m)}(\Lambda)]}{\Lambda}\right) \log \frac{\Lambda}{\Lambda - \lambda^{(M)}\mathbb{E}'[N_\ell^{(\forall m)}(\Lambda)]} - \log 2$$

where inequality (a) is due to that for all $(p,q) \in [0,1]^2, \mathrm{kl}(p,q) \geqslant (1-p)\log(1/(1-q)) - \log 2$.

Notice that the regret attributed to any arm $k$ can be decomposed and lower bounded as follows,

$$R_\ell(\Lambda) \geqslant \sum_{m=1}^{M} N_\ell^{(m)}(\Lambda) \left(\frac{\lambda^{(m)}}{\lambda^{(1)}}\mu_1^{(M)} - \mu_\ell^{(M)}\right),$$

with the constraint in Eq.(24). Since this is a linear programming, we know its solution is reached at its vertex. Therefore, we lower bound the regret as follows,

$$R_\ell(\Lambda) \geqslant \min_{m:\Delta_k^{(m)}>0} \left(\frac{\lambda^{(m)}}{\lambda^{(1)}}\mu_1^{(M)} - \mu_\ell^{(M)}\right) \frac{1}{\mathrm{KL}(v_\ell^{(m)}, v'_\ell^{(m)})}$$

$$\times \left(\left(1 - \frac{\lambda^{(M)}\mathbb{E}[N_\ell^{(\forall m)}(\Lambda)]}{\Lambda}\right) \log \frac{\Lambda}{\Lambda - \lambda^{(M)}\mathbb{E}'[N_\ell^{(\forall m)}(\Lambda)]} - \log 2\right), \tag{25}$$

Notice that the policy is consistent, that is, $\mathbb{E}[N_\ell^{(\forall m)}(\Lambda)] = o(T^a)$ and $\mathbb{E}'[N_k^{(\forall m)}(\Lambda)] = o(\Lambda^a)$ for any $a \in (0,1]$ and any suboptimal arm $k \neq \ell$. We have $\frac{\lambda^{(M)}\mathbb{E}[N_\ell^{(\forall m)}(\Lambda)]}{\Lambda} = o(1)$ and

$$\Lambda - \lambda^{(M)}\mathbb{E}'[N_\ell^{(\forall m)}(\Lambda)] \leqslant \lambda^{(M)}\sum_{k\neq\ell}\mathbb{E}'[N_k^{(\forall m)}(\Lambda)] = o(\Lambda^a).$$

Dividing both sides of Eq.(25) by $\Lambda$, and letting $\Lambda$ go to infinity and $a$ go to 1, we have

$$
\liminf_{\Lambda \to \infty} \frac{\mathbb{E}[R_\ell(\Lambda)]}{\Lambda} \geqslant \min_{m:\Delta_\ell^{(m)}>0} \left( \frac{\lambda^{(m)}}{\lambda^{(1)}} \mu_1^{(M)} - \mu_\ell^{(M)} \right) \frac{1}{\mathrm{KL}(v_\ell^{(m)}, v'^{(m)}_\ell)}
$$

$$
\geqslant \min_{m:\Delta_\ell^{(m)}>0} \left( \frac{\lambda^{(m)}}{\lambda^{(1)}} \mu_1^{(M)} - \mu_\ell^{(M)} \right) \frac{1}{\mathrm{KL}(v_\ell^{(m)}, v_1^{(M)} - \zeta^{(m)} + \epsilon)}
$$

To bound the optimal arm 1's sample cost, we use the same $\nu$ as above and construct another instance $\nu''$. The instance $\nu''$'s reward means are the same to $\nu$ except for arm 1 whose reward means for fidelity $m \in \mathcal{M}_1$ are set as $\mu''^{(m)}_1 = \mu_2^{(m)} + \zeta^{(m)} - \epsilon$. Then, with similar procedure as the above, we obtain

$$
\liminf_{\Lambda \to \infty} \frac{\mathbb{E}[R_1(\Lambda)]}{\Lambda} \geqslant \min_{m:\Delta_k^{(m)}>0} \left( \frac{\lambda^{(m)}}{\lambda^{(1)}} \mu_1^{(M)} - \mu_1^{(M)} \right) \frac{1}{\mathrm{KL}(v_1^{(m)}, v_2^{(m)} + \zeta^{(m)} - \epsilon)}
$$

### F.3 Proof of Theorem 4.8

We first prove a problem dependent regret upper bound as follows and then convert this bound to the problem independent regret upper bound presented in Theorem 4.8.

Denote $\Delta_k^{(M)} = \mu_1^{(M)} - \mu_k^{(M)}$, and, especially, $\Delta_1^{(M)} = 0$.

**Theorem F.1** (Problem-Dependent Regret Upper Bound). *For any $\varepsilon > 0$. Algorithm 3's regret is upper bounded as follows,*

$$
\mathbb{E}[R(\Lambda)] \leqslant \sum_{k:\Delta_k^{(M)}>\varepsilon} \left( \left( \frac{\lambda^{(M)}}{\lambda^{(1)}} \mu_1^{(M)} - \mu_k^{(M)} \right) \left( \frac{16}{(\Delta_k^{(M)})^2} \log \frac{\Lambda(\Delta_k^{(M)})^2}{16\lambda^{(M)}} + \frac{48}{(\Delta_k^{(M)})^2} + 1 \right) + \frac{64}{\Delta_k^{(M)}} \right)
$$

$$
+ \sum_{k:\Delta_k^{(M)}\leqslant\varepsilon} \left( \left( \frac{\lambda^{(M)}}{\lambda^{(1)}} \mu_1^{(M)} - \mu_k^{(M)} \right) \left( \frac{16}{\varepsilon^2} \log \left( \frac{\Lambda\varepsilon^2}{16\lambda^{(M)}} \right) + \frac{32}{3\varepsilon^2} + 1 \right) + \frac{64}{\varepsilon} \right) + \max_{k:\Delta_k^{(M)}\leqslant\varepsilon} \frac{\Lambda}{\lambda^{(1)}} \Delta_k^{(M)}.
$$

$$
\tag{26}
$$

*Proof of Theorem F.1.* By Hoeffding's inequality, we have

$$
\mathbb{P}\left( \hat{\mu}_{k,p}^{(M)} > \mu_k^{(M)} + 2^{-p} \right) \leqslant \exp\left( -\frac{2^{-2p}}{2 \times \frac{1}{2 \times 2^{2p} \log(\Lambda/2^{2p}\lambda^{(M)})}} \right) = \frac{2^{2p}\lambda^{(M)}}{\Lambda}
$$

$$
\mathbb{P}\left( \hat{\mu}_{k,p}^{(M)} < \mu_k^{(M)} - 2^{-p} \right) \leqslant \exp\left( -\frac{2^{-2p}}{2 \times \frac{1}{2 \times 2^{2p} \log(\Lambda/2^{2p}\lambda^{(M)})}} \right) = \frac{2^{2p}\lambda^{(M)}}{\Lambda}.
$$

That is, the empirical mean $\hat{\mu}_k^{(M)}$ is within the confidence interval $(\mu_k^{(M)} - 2^{-p}, \mu_k^{(M)} + 2^{-p})$ with high probability.

Choose any $\varepsilon > \frac{e}{\Lambda}$. Let $\mathcal{K}' = \{k \in \mathcal{K} | \Delta_k^{(M)} > \varepsilon\}$. Denote $p_k := \min\{p : 2^{-p} < \frac{\Delta_k^{(M)}}{2}\}$. From $p_k$'s definition, we have the following inequality

$$
2^{p_k} < \frac{4}{\Delta_k^{(M)}} < 2^{p_k+1}.
\tag{27}
$$

We also note that the cost of pulling a suboptimal arm $k \in \mathcal{K}'$ at highest fidelity $M$ is upper bounded as $\frac{\lambda^{(M)}}{\lambda^{(1)}} \mu_1^{(M)} - \mu_k^{(M)}$, where the factor $\frac{\lambda^{(M)}}{\lambda^{(1)}}$ is because the budget paying to pull an arm at fidelity $M$ can be used to the the arm at fidelity 1 for fractional times.

The rest of this proof consists of two steps. In the first step, we assume that all empirical means are in their corresponding confidence intervals at each phases, and show the algorithm can *properly* eliminate all suboptimal arms in $\mathcal{K}'$—the arm is eliminated in or before the phase $p_k$. In the second step, we upper bound the regret if there are any empirical estimates lying outside their corresponding confidence intervals.

**Step 1.** If all arms' empirical means lie in confidence intervals. That is, any suboptimal arm $k$ is eliminated in or before the phase $p_k$. Because, if $k \in \mathcal{C}_{p_k}$, we have

$$
\hat{\mu}_k^{(M)} \leqslant \mu_k^{(M)} + 2^{-p_k} = \mu_1^{(M)} - \Delta_k^{(M)} + 2^{-p_k}
$$

$$
\overset{(a)}{\leqslant} \mu_1^{(M)} - 4 \times 2^{-p_k-1} + 2^{-p_k} = \mu_1^{(M)} - 2^{-p_k} < \hat{\mu}_1^{(M)} \leqslant \max_{k \in \mathcal{C}_{p_k}} \hat{\mu}_k^{(M)},
$$

where (a) is due to Eq.(27). That is, if this arm $k$ haven't been eliminated before phase $p_k$, it must be eliminated in this phase. Therefore, the total pulling times of this arm $k$ at highest fidelity $M$ is upper bounded as follows,

$$
T_k^{(M)} \leqslant \left\lceil 2^{2p_k} \log \frac{\Lambda}{2^{2p_k}\lambda^{(M)}} \right\rceil \overset{(a)}{\leqslant} \frac{16}{(\Delta_k^{(M)})^2} \log \left( \frac{(\Delta_k^{(M)})^2 \Lambda}{16\lambda^{(M)}} \right) + 1, \tag{28}
$$

where (a) is due to Eq.(27).

We then handle the number of times of pulling arms $k$ with $\Delta_k^{(M)} \leqslant \varepsilon$ in fidelity $M$. Although these arms' total pulling times, eliminated in or before phase $p_k$, are also upper bounded by Eq.(28), their corresponding phases $p_k$ is greater than $\log_2 \frac{2}{\varepsilon}$ and, therefore, cannot be reached. So, these arms' (including the optimal arm 1's) total pulling times in fidelity $M$ is upper bounded by

$$
\frac{16}{\varepsilon^2} \log \left( \frac{\Lambda\varepsilon^2}{16\lambda^{(M)}} \right) + 1.
$$

Therefore, the cost due to pulling arms at fidelity $M$ is upper bounded as follows,

$$
\sum_{k \in \mathcal{K}} \left( \frac{\lambda^{(M)}}{\lambda^{(1)}} \mu_1^{(M)} - \mu_k^{(M)} \right) \left( \frac{16}{\left( \max\left\{ \varepsilon, \Delta_k^{(M)} \right\} \right)^2} \log \left( \frac{\left( \max\left\{ \varepsilon, \Delta_k^{(M)} \right\} \right)^2 \Lambda}{16\lambda^{(M)}} \right) + 1 \right). \tag{29}
$$

After the elimination process, arms with $\Delta_k^{(M)} \leqslant \varepsilon$ may remain in the candidate arm set $\mathcal{C}_p$ and are exploited in turn at fidelity $m = 1$. As some of them are not the optimal arm, this additional cost can be upper bounded as follows,

$$
\max_{k : \Delta_k^{(M)} \leqslant \varepsilon} \frac{\Lambda}{\lambda^{(1)}} \Delta_k^{(M)}. \tag{30}
$$

**Step 2.** There are two cases that some arms are eliminated improperly: for an suboptimal arm $k$, either

2.1 The suboptimal arm $k$ is *not* eliminated in (or before) the phase $p_k$, and the optimal arm 1 is in $\mathcal{C}_{p_k}$ in phase $p_k$; or

2.2 The suboptimal arm $k$ is eliminated in (or before) the phase $p_k$, and the optimal arm 1 is *not* in $\mathcal{C}_{p_k}$ in phase $p_k$.

Case 2.1's happening means that arm $k$ is not eliminated in or before phase $p_k$, which can only happen when the arm's empirical mean $\hat{\mu}_k^{(M)}$ lies outside its corresponding confidence interval. This event in or before phase $p_k$ is with a probability no greater than $2 \times \frac{2^{2p_k}\lambda^{(M)}}{\Lambda}$. Since this event may happen to any suboptimal arm $k \in \mathcal{K}'$, then the regret of this case is upper bounded by

$$
\sum_{k \in \mathcal{K}'} \frac{2^{2p_k+1}\lambda^{(M)}}{\Lambda} \frac{\Lambda}{\lambda^{(M)}} \left( \frac{\lambda^{(M)}}{\lambda^{(1)}} \mu_1^{(M)} - \mu_k^{(M)} \right) = \sum_{k \in \mathcal{K}'} 2^{2p_k+1} \left( \frac{\lambda^{(M)}}{\lambda^{(1)}} \mu_1^{(M)} - \mu_k^{(M)} \right)
$$

$$
\overset{(a)}{\leqslant} \sum_{k \in \mathcal{K}'} \frac{32}{(\Delta_k^{(M)})^2} \left( \frac{\lambda^{(M)}}{\lambda^{(1)}} \mu_1^{(M)} - \mu_k^{(M)} \right), \tag{31}
$$

where (a) is due to Eq.(27).

If Case 2.2 happens, the optimal arm 1 is not in the candidate arm set $\mathcal{C}_{p_k}$ in phase $p_k$. We denote $p_1$ as the phase that the optimal arm 1 is eliminated, and it is at this phase that some arms' empirical means lie outside their confidence interval so that this mis-elimination happens. The probability of this event is upper bounded by $2 \times \frac{2^{2p_1}\lambda^{(M)}}{\Lambda}$. We assume that arms $k$ with $p_k < p_1$ are eliminated in or before phase $p_i$ properly; otherwise, the regret is counted in Case 2.1. Therefore, the optimal arm 1 eliminated in phase $p_1$ should be eliminated by an arm $k$ with $p_k \geqslant p_1$. Consequently, the maximal per time slot regret in Case 2.2 is among arms with $p_k \geqslant p_1$. Denote $p_\varepsilon := \min\{p|2^{-p} < \frac{\varepsilon}{2}\}$. We bound the cost of Case 2.2 as follows,

$$\sum_{p_1=0}^{\max_{i\in\mathcal{K}'} p_i} \sum_{k>1:p_k\geqslant p_1} \frac{2^{2p_1+1}\lambda^{(M)}}{\Lambda}\frac{\Lambda}{\lambda^{(M)}} \cdot \max_{k'>1:p_{k'}\geqslant p_1} \left(\frac{\lambda^{(M)}}{\lambda^{(1)}}\mu_1^{(M)} - \mu_{k'}^{(M)}\right)$$

$$= \sum_{p_1=0}^{\max_{i\in\mathcal{K}'} p_i} \sum_{k>1:p_k\geqslant p_1} 2^{2p_1+1} \cdot \left(\frac{\lambda^{(M)}}{\lambda^{(1)}}\mu_1^{(M)} - \mu_1^{(M)} + \max_{k'>1:p_{k'}\geqslant p_1} \Delta_{k'}^{(M)}\right)$$

$$\overset{(a)}{\leqslant} \sum_{p_1=0}^{\max_{i\in\mathcal{K}'} p_i} \sum_{k>1:p_k\geqslant p_1} 2^{2p_1+1} \cdot \left(\frac{\lambda^{(M)}}{\lambda^{(1)}}\mu_1^{(M)} - \mu_1^{(M)} + 4 \times 2^{-p_1}\right)$$

$$\overset{(b)}{\leqslant} \sum_{k>1} \sum_{p_1=0}^{\min\{p_k,p_\varepsilon\}} 2^{2p_1+1} \cdot \left(\frac{\lambda^{(M)}}{\lambda^{(1)}}\mu_1^{(M)} - \mu_1^{(M)} + 4 \times 2^{-p_1}\right)$$

$$\leqslant \sum_{k>1} \left(\left(\frac{\lambda^{(M)}}{\lambda^{(1)}}\mu_1^{(M)} - \mu_1^{(M)}\right) \sum_{p_1=0}^{\min\{p_k,p_\varepsilon\}} 2^{2p_1+1} + \sum_{p_1=0}^{\min\{p_k,p_\varepsilon\}} 2^{p_1+3}\right)$$

$$\leqslant \sum_{k>1} \left(\left(\frac{\lambda^{(M)}}{\lambda^{(1)}}\mu_1^{(M)} - \mu_1^{(M)}\right) \frac{2^{2\min\{p_k,p_\varepsilon\}+1}}{3} + 2^{\min\{p_k,p_\varepsilon\}+4}\right)$$

$$\leqslant \sum_{k\in\mathcal{K}'} \left(\left(\frac{\lambda^{(M)}}{\lambda^{(1)}}\mu_1^{(M)} - \mu_1^{(M)}\right) \frac{2^{2p_k+1}}{3} + 2^{p_k+4}\right) + \sum_{k>1,k\notin\mathcal{K}'} \left(\left(\frac{\lambda^{(M)}}{\lambda^{(1)}}\mu_1^{(M)} - \mu_1^{(M)}\right) \frac{2^{2p_\varepsilon+1}}{3} + 2^{p_\varepsilon+4}\right)$$

$$\overset{(c)}{\leqslant} \sum_{k\in\mathcal{K}'} \left(\left(\frac{\lambda^{(M)}}{\lambda^{(1)}}\mu_1^{(M)} - \mu_1^{(M)}\right) \frac{32}{3(\Delta_k^{(M)})^2} + \frac{64}{\Delta_k^{(M)}}\right) + \sum_{k>1,k\notin\mathcal{K}'} \left(\left(\frac{\lambda^{(M)}}{\lambda^{(1)}}\mu_1^{(M)} - \mu_1^{(M)}\right) \frac{32}{3\varepsilon^2} + \frac{64}{\varepsilon}\right)$$

$$\leqslant \sum_{k\in\mathcal{K}'} \left(\left(\frac{\lambda^{(M)}}{\lambda^{(1)}}\mu_1^{(M)} - \mu_k^{(M)}\right) \frac{32}{3(\Delta_k^{(M)})^2} + \frac{64}{\Delta_k^{(M)}}\right) + \sum_{k>1,k\notin\mathcal{K}'} \left(\left(\frac{\lambda^{(M)}}{\lambda^{(1)}}\mu_1^{(M)} - \mu_k^{(M)}\right) \frac{32}{3\varepsilon^2} + \frac{64}{\varepsilon}\right)$$

$$\tag{32}$$

where (a) and (c) are due to Eq.(27), and (b) is due to the property of swapping two summations.

Summing up the costs in Eq.(29), Eq.(30), Eq.(31), and Eq.(32) concludes the proof. $\qquad\square$

Next, we derive the problem-independent regret bound from Eq.(26). When $\Lambda$ is large, the $O(\log\Lambda)$ and $O(\Lambda)$ terms dominate other terms in Eq.(26). For any given $\varepsilon$, if $\Lambda$ is large enough, we always have $\varepsilon > 4\sqrt{\frac{\lambda^{(M)}e}{\Lambda}}$, which, with some calculus, guarantees that $\frac{16}{(\Delta_k^{(M)})^2}\log\frac{\Lambda(\Delta_k^{(M)})^2}{16\lambda^{(M)}} < \frac{16}{\varepsilon^2}\log\frac{\Lambda\varepsilon^2}{16\lambda^{(M)}}$ for all $\Delta_k^{(M)} > \varepsilon$. Therefore, we can scale all logarithmic arm pulling times as $\frac{16}{\varepsilon^2}\log\frac{\Lambda\varepsilon^2}{16\lambda^{(M)}}$, and upper bound the pulling cost $\left(\frac{\lambda^{(M)}}{\lambda^{(1)}}\mu_1^{(M)} - \mu_k^{(M)}\right)$ by $\frac{\lambda^{(M)}}{\lambda^{(1)}}\mu_1^{(M)}$. We derive the problem-independent regret upper bound as follows,

$$\mathbb{E}\left[R(\Lambda)\right] \leqslant \frac{K\mu_1^{(M)}\cdot\lambda^{(M)}}{\lambda^{(1)}}\frac{16}{\varepsilon^2}\log\frac{\Lambda\varepsilon^2}{16\lambda^{(M)}} + \frac{\Lambda}{\lambda^{(1)}}\varepsilon$$

$$\leqslant \frac{K\mu_1^{(M)}\cdot\lambda^{(M)}}{\lambda^{(1)}}\frac{16}{\varepsilon^2}\log\frac{\Lambda}{16\lambda^{(M)}} + \frac{\Lambda}{\lambda^{(1)}}\varepsilon$$

$$\overset{(a)}{\leqslant} 2\left(\frac{16K\mu_1^{(M)}\lambda^{(M)}}{\lambda^{(1)}}\log\frac{\Lambda}{16\lambda^{(M)}}\right)^{\frac{1}{3}}\left(\frac{\Lambda}{\lambda^{(1)}}\right)^{\frac{2}{3}},$$

where the equation of (a) holds when $\varepsilon = \left( \frac{K\mu_1^{(M)} \log(\Lambda/16\lambda^{(M)})}{(\Lambda/16\lambda^{(M)})} \right)^{\frac{1}{3}}$.

