# OpenReview forum: "Multi-Fidelity Multi-Armed Bandits Revisited"
_NeurIPS.cc/2023/Conference — NeurIPS 2023 poster_

### Official Review · Reviewer_3tC7 · 2023-07-05

**Soundness:** 3 good
**Presentation:** 3 good
**Contribution:** 3 good
**Rating:** 6
**Confidence:** 3

**Summary:**

The authors consider the problem of multi-fidelity multi-armed bandits under fixed-confidence BAI and regret-minimization objectives. The best-arm identification algorithm is based on lower-upper confidence bound and associated cost-complexity.  A novel definition of regret is introduced which captures the fact that fidelities influence the accuracy of affected rewards. Lower and upper bounds are provided for both problem cases.

**Strengths:**

This paper provides a novel outlook at the multi-fidelity MAB problem introduced by Kandaswamy et. al. The BAI algorithm is an index-based algorithm with a new UCB based procedure to determine the optimal fidelity to be sampled.

**Weaknesses:**

-

**Questions:**

-

**Limitations:**

-

---

> ### Author Rebuttal · Authors · 2023-08-10
>
> We thank the review for providing positive feedback to
> this work.

---

### Official Review · Reviewer_zHKa · 2023-07-07

**Soundness:** 4 excellent
**Presentation:** 3 good
**Contribution:** 3 good
**Rating:** 7
**Confidence:** 4

**Summary:**

This paper studies the multi-fidelity multi-armed bandits problem where each arm has different fidelities and observation accuracy. There are many existing works in this problem. This paper studies both best arm identification with fixed confidence and regret minimization. There are some novel theoretical results in both problems.

**Strengths:**

1. This paper studies an important problem and overall presentation is good so it’s an easy-to-follow paper.
2. This paper claims to be the first work to study the BAI task under the MF-MAB model.
3. It’s great to see many application-oriented discussion after each theoretical analysis.
4. The tightness of theoretical results is checked; see Remark 3.6, 4.4, 4.5.
5. Compared with Kandasamy et al. [19], the cost complexity definition in this paper makes more sense to me.

**Weaknesses:**

1. Although this paper claims to be the first work to study the BAI task under the MF-MAB model, I do have a question on comparison to Kandasamy et al. [18] as discussed in line 76-82. Yes, simple regret minimization in continuous domain is equivalent to BAI in discrete domain, but after discretization of continuous domain how is the algorithm in this paper compared to Kandasamy et al. [18]?
2. In Remark 3.8, I don’t think $\mu_1$ and $\mu_2$ can be easily obtained in real-world applications. Yes, a perfect classification means $\mu=1.0$ but that’s because of the output range. For the 2nd-best arm, its \mu is hard to get.
3. A minor comment. I do like application-oriented discussion but why do they appear after each technical stuff? I think first being motived by applications, stating the problem, and presenting the results makes more sense to me.
4. A minor suggest. Instead of denoting best arm as arm 1 and 2nd-best arm as arm 2, $a_1$ and $a_2$ look better.

**Questions:**

1. In line 225, it’s good to compare explore-a vs. explore-b but what’s the criterion to select one of them to perform best?
2. There are some technical terms that need more explanation.
(2.1) In line 109, why the error upper bounds can be revealed to the learning agent? Could you give a real-world example and explain what are they in this example?
(2.2) In line 131, are v^(m)_k different according to different algorithms? If yes, why can they appear in lower bound?
(2.3) In line 172, \beta seems like something similar as in GP-UCB paper. Could you explain why \beta is defined as that? Pointing it to somewhere in the proof also works.
(2.4) In line 182, how “half” is chosen? I guess any constant between 0 and 1 works, right? If a constant C is used instead of half, how does it affect theoretical results?

**Limitations:**

Limitations are discussed in last section.

---

> ### Author Rebuttal · Authors · 2023-08-10
>
> We highly appreciate the reviewer for these instructive comments. Below, we answer weaknesses and questions point by point according to the reviewer's numerical label.
>
> **Weaknesses**
> > *Although this paper claims to be the first work to study the BAI task under the MF-MAB model, I do have a question on comparison to Kandasamy et al. [18] ...*
> 1. Under the MF-MAB model, we are claiming to be the first to study the best arm identification *with fixed confidence* setting, which is different from best arm identification *with fixed budget* setting---the objective of Kandasamy et al. [18] after domain discretization. The fixed confidence refers to give a fixed confidence parameter $\delta$, to find the best arm with as small costs as possible, where the *cost complexity* is the minimizing objective, while the fixed budget refers to give fixed cost as budget, to find the best arm with confidence as high as possible, where the *fail probability* is the minimizing objective. With two different minimization objectives, both settings and their algorithms are incomparable.
>
> > *In Remark 3.8, I don't think $\mu_1$ and $\mu_2$ can be easily obtained in real-world...*
>
> 2. We note that to run our algorithm, it is not necessary to get the true means of top two arms. The upper bound of $\mu_1^{(M)}$ and lower bound of $\mu_2^{(M)}$ are enough to make sure the algorithm works properly (see Line 187). In classification scenario mentioned by the reviewer, classification accuracy of the model with the default hyperparameter setting could be considered a safe (and reasonably tight) lower bound for the 2nd-best arm. With additional application-based information, e.g., in the hyperparameter optimization example discussed in Remark 3.8, one can have better bounds for these two reward means.
>
> 3. We highly appreciate this suggestion. We will move Section 3.4 for application discussion to the beginning of Section 3 to better motivate the multi-fidelity best arm identification task.
>
> 4. We appreciate the reviewer for this suggestion. In the final version of this paper, we will update the notations of arm index and means accordingly.
>
> **Questions**
>
> > *1. In line 225, it's good to compare explore-a vs. explore-b but what's the criterion to select one of them to perform best?*
>
> -   Theoretically, For when do we prefer $\texttt{Explore-A}$ than $\texttt{Explore-B}$, one can informally say if $\tilde{G}$ is small ($\tilde{G}=O(M\tilde{H})$), $\texttt{Explore-A}$ is preferred, and, otherwise if $\tilde{G}$ is large, $\texttt{Explore-B}$ is preferred.
>
> - Practically, we can just run both $\texttt{Explore-A}$ and $\texttt{-B}$ in parallel for each arm $k$. If $\texttt{Explore-A}$ starts to converge on a fidelity, i.e., a large proportion of pull times on this arm is at this fidelity, we stop $\texttt{Explore-B}$ and continue $\texttt{Explore-A}$. Otherwise, if $\texttt{Explore-B}$ commits to fidelity at first, we stop $\texttt{Explore-A}$ and follow $\texttt{Explore-B}$ to commit this fidelity. With this heuristic approach, we can expect the empirical cost to be close to the smaller one of running either $\texttt{Explore-A}$ or $\texttt{-B}$ alone.
>
> > *2. There are some technical terms that need more explanation...*
>
> -  2.1 Due to the limited space in rebuttal, we kindly refer the reviewer to [the first reply to review rkrs](https://openreview.net/forum?id=oi45JlpSOT&noteId=SfykMGfX3s) where we elaborate on why the error upper bounds can be revealed in hyperparameter optimization.
>
> -  2.2 We note that the distribution $\nu^{(m)}_k$ is the reward distribution of pulling arm $k$ at fidelity $m$. It depends only on the multi-fidelity multi-armed bandits model parameters, instead of any specific algorithm.
>
> - 2.3 Yes, the $\beta$ is the confidence radius for constructing the upper confidence bound (UCB), and there is also a corresponding term in GP-UCB. Recall $$\beta(n,t,\delta) = \sqrt{\frac{\log(Lt^4/\delta)}{n}},$$ where $n$ is the number of observations, $t$ is the current time slot, $\delta$ is the confidence parameter, and $L\,(>0)$ is a constant factor. With Hoeffding's inequality, one usually define the confidence radius as $\sqrt{\frac{\log(1/\delta)}{n}}$ to guarantee that the true mean $\mu$ is insides the interval $(\hat{\mu} - \sqrt{\frac{\log(1/\delta)}{n}}, \hat{\mu} + \sqrt{\frac{\log(1/\delta)}{n}})$ with a confidence $1-\delta$. The reason why we replace $1/\delta$ with $Lt^4/\delta$ --- i.e., increase the confidence to $1 - \frac{\delta}{Lt^4}$ --- is because in the proof, we need to utilize the confidence intervals many times, e.g., for many time slots and for all arms and fidelities, and thus we need a higher the confidence for each of these intervals so that their union confidence is high enough for our fixed confidence theoretical guarantee.
>
> -   2.4 Yes, the selection of "half" is just for the simplicity of presentation, and any constant between $(0,1)$ would work. If a constant $C\in (0,1)$ is used, then the factor $3$ of
>     right-hand side in the condition of Line 10 of $\texttt{Explore-B}$ becomes $\frac{1+C}{1-C}$, and there would be an additional $(C^{-2} + (1-C)^{-2})$ multiplier appearing in the cost complexity upper bound of [Explore-B]{.smallcaps} in Eq. (8). In the $(C^{-2} + (1-C)^{-2})$ multiplier, the first term $C^{-2}$ factor is because a small $C$ implies that $\texttt{Explore-B}$ may commit to a very inefficient suboptimal fidelity and thus suffer a high cost, while the second term $(1-C)^{-2}$ corresponds to that if $C$ is close to $1$, then it becomes very difficult for $\texttt{Explore-B}$ to find a good fidelity to commit, and thus $\texttt{Explore-B}$ has to spend a large cost in exploration.

---

> > ### Comment · Reviewer_zHKa · 2023-08-18
> > **Thanks for rebuttal**
> >
> > Authors' rebuttal addresses most of my concerns and I really encourage authors to polish the paper presentation by incorporating our discuss in review and rebuttal in the next version. I'm happy to increase my rating from 6 to 7.

---

### Official Review · Reviewer_Rfmo · 2023-07-11

**Soundness:** 3 good
**Presentation:** 3 good
**Contribution:** 2 fair
**Rating:** 4
**Confidence:** 2

**Summary:**

The paper studies the multi-fidelity multi-armed bandit problem where each arm can be pulled at different fidelities, providing better or worse estimate of the true mean, at a different cost. The paper studies best arm identification with fixed confidence, provides a lower bound on the cost complexity, and an algorithm with cost complexity upper bounds. In addition, the paper proposes a new regret definition which finds multiple applications. Upper and lower bounds on the proposed regret are proved for the instance dependent and instance independent cases.

**Strengths:**

- The paper studies a practical problem that has various application.
- It is well-written and easy to follow.
- The proposed definition of regret looks natural to study and has multiple applications.
- The paper provides upper and lower bounds that are tight, in some cases, for both best arm identification and regret minimization.

**Weaknesses:**

My main concerns are the following:

- The proposed algorithm for BAI requires the estimates $\tilde{\mu}_1^{(M)}, \tilde{\mu}_2^{(M)}$ which is not a realistic assumption. Even though the author argue that this is possible in some applications, it is still unrealistic for many other applications. These assumptions make the problem theoretically and practically less interesting.
- The cost upper bound for BAI depends on $\tilde{m}^\star_k$, which depend on $\tilde{\mu}_1^{(M)}, \tilde{\mu}_2^{(M)}$ that are not guaranteed to be tight estimates. It also has extra terms that are not matched in the lower bound.

- The problem dependent regret upper bound is not tight in most cases.

**Questions:**

- Can the proposed regret be upper bounded by $\lambda^{(M)}$ times simple regret considered in the literature? How does the proposed upper bound compare to this?

- Please see weaknesses.

**Limitations:**

Yes.

---

> ### Author Rebuttal · Authors · 2023-08-10
>
> >  - *The proposed algorithm for BAI requires the estimates
>     $\tilde{\mu}\_1\^{(M)}, \tilde{\mu}\_2\^{(M)}$ which is not a realistic
>     assumption. ...*
> > - *The cost upper bound for BAI depends on
>     $\tilde{m}\_k\^{\star}$ , which depend on
>     $\tilde{\mu}\_1\^{(M)}, \tilde{\mu}\_2\^{(M)}$ that are not guaranteed
>     to be tight estimates. It also has extra terms that are not matched
>     in the lower bound.*
>
> We thank the reviewer for pointing this out. We
>     acknowledge that when the reward mean bounds for the top two arms
>     are loose, the cost complexity would have extra terms. We notice
>     that practically, there are applications, e.g., the hyperparameter
>     optimization example in
>     Remark 3.8, that provide tight bounds for top two
>     arms.
>
> > *The problem-dependent regret upper bound is not tight
>     in most cases.*
>
> We acknowledge the problem-dependent regret upper
>     and lower bounds only match in a class of instances mentioned in
>     Remark 4.5, where we also illustrate that this class of instance,
>     though not general, is common in practice. On the other hand, we
>     notice that our algorithm's problem-independent regret upper bound
>     matches the lower bound up to logarithmic factor, and often in
>     bandits literature, if one algorithm is good at problem-independent
>     case, it would be slightly worse in the problem-dependent case.
>     Therefore, since our algorithm's problem-independent bound is tight,
>     the result that its problem-dependent upper bound is not very tight
>     is expected.
>
> > *Can the proposed regret be upper bounded by
>     $\lambda^{(M)}$ times simple regret considered in the literature?
>     How does the proposed upper bound compare to this?*
>
> Recall that the simple regret in the literature refers
>     to the single reward mean difference between the optimal arm and the
>     final output arm by their concerned algorithm, while our regret is
>     the cumulative reward mean differences between the optimal arm and
>     the pulled arm by the concerned algorithm over all decision rounds.
>     Hence, with different base units, $\lambda^{(M)}$ times their simple
>     regret is incomparable to our regret upper bound. Although one can
>     compare our regret upper bound with simple regret in prior works,
>     e.g.,  Kandasamy et al. [19], via multiplying their simple regret
>     upper bound by the time horizon $T$, this is rather unfair. Because
>     our regret upper bound counts the cost when the learning was not
>     accurate at the beginning, while simply times $T$ with simple regret
>     enjoys the best accurate over all decision rounds.
>
> - [19] Kirthevasan Kandasamy, Gautam Dasarathy, Barnabas Poczos, and Jeff Schneider. The multifidelity multi-armed bandit. Advances in neural information processing systems, 29, 2016.

---

> ### Author Response · Authors · 2023-08-21
>
> This is a gentle reminder for the reviewer that the author-reviewer discussion will end in less than 15 hours. In case you have any further questions, please feel free to ask; we are happy to answer them.

---

### Official Review · Reviewer_QHcL · 2023-07-21

**Soundness:** 3 good
**Presentation:** 3 good
**Contribution:** 3 good
**Rating:** 5
**Confidence:** 2

**Summary:**

This paper studys the problem of Multi-Fidelity Multi-Armed Bandits (MF-MAB) where each arm can be pulled at different fidelity with different rewards and costs. The main contribution of this paper includes derive the cost complxity lower bound for best arm idenfication  with fixed confidence and a new definition of regret for regret minimization.

**Strengths:**

The theorical results about the cost complexity lower bound is interesting and the proof of each theorem is comprehensive.

**Weaknesses:**

- In Algorithm 1, the best arm and second best arm is selected according to the up confidence bound (UCB).
    - Why use the UCB given that UCB of an arm is larger than that of another arm does not necessary mean that the first arm is better? How about use LCB?
    - Since the UCB and LCB both depends on the confidence radius, is it possible that the algorithm will come to an earlier decision to pick the arm that is not well explored? Here, I am assuming that if the arm is not well expored the confidence radius will be large thus the UCB will also be large.

- It would be better to have some toy examples/experiments to show the effectiveness of the theorical results?

**Questions:**

As in Weakness section

**Limitations:**

Yes, the authors addressed the limitations.

---

> ### Author Rebuttal · Authors · 2023-08-10
>
> > *Why use the UCB given that UCB of an arm is
>         larger than that of another arm does not necessary mean that the
>         first arm is better? How about use LCB?*
>
>  We believe this question is about why we use UCB
>         indices in Line 4 of Algoithm 1 (LUCB framework) to pick top two
>         arms and why not use LCB indices. We note that arms with high
>         uncertainty (e.g., lack exploration) would have a large
>         confidence raduis (see $\beta(n,t,\delta)$ in Line 172) and thus
>         a large UCB. As the algorithm is looking for top two arms,
>         picking arms with high UCB can direct the algorithm to explore
>         these under-explored arms so as to balance the exploration of
>         all arms. In constrast, LCB would mislead this exploration away
>         from under-explored arms: the less one arm is explore, the lower
>         its LCB would be, and, via picking arms with high LCB, there
>         would be less chance to explore the under-explored arms.
>
> > *Since the UCB and LCB both depends on the
>         confidence radius, is it possible that the algorithm will come
>         to an earlier decision to pick the arm that is not well
>         explored? Here, I am assuming that if the arm is not well
>         expored the confidence radius will be large thus the UCB will
>         also be large.*
>
> As long as the interval composed by UCB and LCB of
>         an estimate contains the true reward mean --- which holds with a
>         high confidence, the algorithm would pick the right optimal arm
>         for sure; even when the confidence radius is large. This is
>         because when the condition in Line 3 of LUCB algorithm does not
>         hold, i.e., the arm $\ell_t$ (with highest UCB index)'s LCB is
>         greater than the second-largest UCB index, we have that
>         $$\mu\_{\ell_t} \overset{(a)}> \text{LCB}\_{\ell_t}
>                                 \overset{(b)}> \text{UCB}\_{k} \overset{(c)}> \mu_k, \text{ for any arm }k\neq \ell_t,$$
>         where inequality (a) and (c) is because the confidence interval
>         contains the true reward mean, and inequality (b) is because
>         Line 3 of Algorithm 1 does not hold. That is, the arm $\ell_t$
>         is indeed the best arm, and the algorithm can identify it as the
>         best arm with high confidence.
>
> > *It would be better to have some toy
>     examples/experiments to show the effectiveness of the theoretical
>     results?*
>
> We are running more experiments, in particular comparing to the baseline LUCB on the highest fidelity, and we hope the experiments will conclude in a few days. Once it finishes we will post it as a reply. For now, please refer to Figure 1 for an empirical comparison of our proposed two exploration policies.

---

> ### Author Response · Authors · 2023-08-20
> **New experiment updated in general comment**
>
> This is a gentle reminder to the reviewer that we added [a new simulation in the general comments](https://openreview.net/forum?id=oi45JlpSOT&noteId=qMzrbHt8FZ). It compares our two exploration policies to one baseline and illustrates when our exploration policies (utilizing low fidelity) outperform the baseline (not utilizing low fidelity).

---

### Official Review · Reviewer_ZToP · 2023-07-26

**Soundness:** 4 excellent
**Presentation:** 3 good
**Contribution:** 3 good
**Rating:** 7
**Confidence:** 3

**Summary:**

In this work authors revisit the multi-fidelity multi armed bandits and provide results i.e., upper and lower bounds for cost complexity for best arm identification with fixed confidence objective using the Lower-upper confidence bound framework. Further, this work introduces 3 procedures for finding optimal fidelity where the comparable upper bound results are given for 1st and 2nd procedure. Finally, the regret minimization objective is discusses briefly, introducing new definition of regret which differs from the definition introduced in the prior work.

**Strengths:**

The paper has made a significant effort to keep clear problem framework and analysis. The proposed method is novel for the best arm identification objective. Further, a detailed theoretical analysis of LUCB framework for both best arm identification objective and regret minimization objective, and comparable upper and lower bounds on the cost complexity and regret respectively are theoretically sound and well presented.

**Weaknesses:**

This work assumes the reward distributions are on bounded support which may not be the case always. Further the knowledge of upper and lower bounds of 1st and 2nd arm respectively is assumed and when these bounds are too loose then the upper bounds on the cost complexity would not be comparable to lower bounds. The assumption 3.3 on which all the results are dependent may not hold always. The paper is missing the experimental evaluation of the proposed framework and comparison with the existing methods. Lastly, some times paper is hard to follow when 1st and 2nd arms are mentioned as to when is the ground truth being referred and when is the indexing of arms being referred.

**Questions:**

Is the assumption 3.3 expected to hold in general?

---

> ### Author Rebuttal · Authors · 2023-08-10
>
> > *This work assumes the reward distributions are on
>     bounded support which may not be the case always.*
>
> Assuming the reward distribution is bounded is common
>     and standard in bandits literature [@auer2002finite]. With known
>     approaches in bandits, this assumption can be easily extended to a
>     more general sub-gaussian one [@lattimore2020bandit §5.3], and, with
>     some efforts, this assumption can be relaxed to heavy tail
>     distribution [@bubeck2013bandits].
>
> - auer2002finite: Peter Auer, Nicolo Cesa-Bianchi, and Paul Fischer. Finite-time analysis of the multiarmed
> bandit problem. Machine learning, 47:235–256, 2002
> - lattimore2020bandit: Tor Lattimore and Csaba Szepesvári. Bandit algorithms. Cambridge University Press, 2020.
> - bubeck2013bandits: Sébastien Bubeck, Nicolo Cesa-Bianchi, and Gábor Lugosi. Bandits with heavy tail. IEEE
> Transactions on Information Theory, 59(11):7711–7717, 2013.
>
> >  *Further the knowledge of upper and lower bounds of
>     1st and 2nd arm respectively is assumed ...*
>
> We acknowledge that when the reward mean bounds for
>     the top two arms are loose, the cost complexity would also be loose.
>     We notice that practically, there are applications, e.g., the
>     hyperparameter optimization example in
>     Remark 3.8, that provides tight reward mean bounds
>     for the top two arms.
>
> > *The assumption 3.3 on which all the results are
>     dependent may not hold always.*
>
> First, we kindly remind the reviewer that only Theorem
>     3.4 relies on Assumption 3.3. All other theoretical results, e.g.,
>     cost complexity lower bound and regret minimization bounds do not
>     need this assumption.
>
> More importantly, we notice that Assumption 3.3 can be removed
>     by slightly changing the definition of value $c$ in the proof of
>     Theorem 3.4 at
>     Appendix C.2 as
>     $$c = \frac{(\mu_1 + \max_k (\mu_k^{(m_k^*)} + \zeta^{(m_k^*)}))}{2}.$$
>     With this new $c$, the inequality (a) in Step 2 and inequality (a)
>     in Step 3 --- where
>     Assumption 3.3 were used --- can go through without
>     the assumption.
>
> > *The paper is missing the experimental evaluation of
>     the proposed framework and comparison with the existing methods.*
>
> We thank the reviewer for giving the suggestion. We are running more experiments, in particular comparing to the baseline LUCB on the highest fidelity, and we hope the experiments will conclude in a few days. Once it finishes we will post it as a reply. For now, please refer to Figure 1 for an empirical comparison of our proposed two exploration policies.
>
> > *Lastly, some times paper is hard to follow when 1st
>     and 2nd arms are mentioned as to when is the ground truth being
>     referred and when is the indexing of arms being referred.*
>
> We thank the reviewer for pointing out this confusion.
>     As another reviewer zHKa suggested, in the final version of this
>     paper, we will use $a_1, a_2$ and $\tilde{a}_1,\tilde{a}_2$ to refer
>     the ground truth and estimated top arm indexes respectively, and use
>     $\mu_1^{(m)}, \mu_2^{(m)}$ and
>     $\tilde{\mu}_1^{(m)},\tilde{\mu}_2^{(m)}$ explicitly for reward
>     means.
>
> > *Is the assumption 3.3 expected to hold in general?*
>
> Please refer to our reply above. The assumption 3.3
>     can be removed by slightly changing our proof.

---

> > ### Comment · Reviewer_ZToP · 2023-08-18
> >
> > I thank the authors for the response. The reviewer has no further questions.

---

### Official Review · Reviewer_rkrs · 2023-07-27

**Soundness:** 2 fair
**Presentation:** 1 poor
**Contribution:** 2 fair
**Rating:** 3
**Confidence:** 4

**Summary:**

This paper studies the the multi-fidelity multi-armed bandit setting where each arm is associated with a cost (fidelity) and observed reward. In this setting when the learner pulls an arm $k \in \mathcal{K}:=\{1, \ldots, K\}$ at fidelity $m \in \mathcal{M}:=\{1, \ldots, M\}$, it pays a cost of $\lambda^{(m)}$ and observes a reward $X_k^{(m)}$ whose mean $\mu_k^{(m)}$ is not too far away from the arm's true reward mean $\mu_k$. This paper studies both the best arm identification and regret minimization setting. For the best arm setting, they propose a LUCB algorithm where the key novelty/challenge (different from standard MAB) is to decide the fidelities for exploring the critical arms. This fidelity selection faces an accuracy-cost trade-off, i.e., higher fidelity (accuracy) but suffering higher cost, or lower fidelity (accuracy) but enjoying lower cost. Hence, this trade-off is different from the common exploration-exploitation trade-off in classic bandits. To address this trade-off in fidelity selections, they propose a UCB-type policy which is called upon by the LUCB algorithm that finds the optimal fidelity $m_k^*$ in (3) for each arm $k$ (EXPLORE-A) and an explore-then-commit policy stopping at a good fidelity that is at least half as good as the optimal one (EXPLORE-B). In theorem 3.4 they show that the sample complexity upper bound matches the lower bound upto constant factors. they also study the regret minimization setting.

**Strengths:**

1) The best arm identification in multi-fidelity multi-armed bandit setting is novel and interesting setting.
2) The proposed solution seems novel and is theoretically analyzed.
3) The sample complexity upper bound matches the lower bound under certain assumptions.

**Weaknesses:**

1) You assume that the reward distribution and the mean $\mu_k^{(m)}$ of each arm $k$ at fidelity $m$ are unknown, while the costs $\lambda^{(m)}$ 's and the error upper bounds $\zeta^{(m)}$ 's are known to the learning agent. This setting assumption seems very contrived.  Is this a standard assumption? Can you point out some references? Can you give some motivating examples where this setting assumption makes sense.

2) The writing needs to substantially improve. There are too many results packed into the paper, without discussion on a specific result in depth. For example one of the main contribution is the regret bound, however, the regret bound theorem is moved to appendix D entirely. Also the regret minimization algorithm is not presented in the main paper and is presented in Appendix D. The novelty of the arm elimination algorithm is not clear to me. Is the regret bound of the regret minimization algorithm tight?

3) This is mainly a theory paper, yet the technical novelty of your approach is not clear to me. What are the technical challenges in proving Theorem 3.4? I think this needs a detailed discussion. For example the [Kandasamy et al.](https://papers.nips.cc/paper_files/paper/2016/file/2ba596643cbbbc20318224181fa46b28-Paper.pdf) paper tackles the multi-fidelity setting for regret minimization and [Kauffmann et al](http://proceedings.mlr.press/v30/Kaufmann13.pdf) studies the LUCB algorithm for single fidelity. My guess is that the proof of thorem 3.4 must use these papers. Can the authors briefly discuss the proof technique.

4) There is no experiment in the main paper. Yet, multi-fidelity paper do substantial experiments on their setting (See [Kandasamy et al.](https://papers.nips.cc/paper_files/paper/2016/file/2ba596643cbbbc20318224181fa46b28-Paper.pdf) ). What are the baselines in your experiments?

5) There is a summation over $m\in\mathcal{M}$ in the Explore-B sample complexity bound, which is not present in Explore-A bound. It is said that when $\tilde{G}=O(M \tilde{H})$, the cost complexity upper bound of EXPLORE-A is less than that of EXPLORE-B. However, if $\tilde{G}=O(M \tilde{H})$ then EXPLORE-A bound is far away from the lower bound. Can you clarify this? Also what are the implications of $\tilde{G}=O(M \tilde{H})$ and what settings have these? In short, when do we prefer EXPLORE-A than EXPLORE-B?

**Questions:**

See weakness section.

**Limitations:**

See weakness section.

---

> ### Author Rebuttal · Authors · 2023-08-10
>
> We appreciate the reviewer for spending time reviewing this paper. Below,  we answer weakness point by point according to numerical order.
>
> 1. Yes, this is a standard setting of multi-fidelity multi-armed bandits proposed by Kandasamy et al. [18, 20].
>
>     **On known error upper bound** We present the hyperparameter optimization as a motivating example in Section 3.4, where arms correspond to different groups of hyperparameters and fidelities correspond to the different amounts of training resources for testing a group of hyperparameters. There are benchmarks for multi-fidelity hyperparameter optimization, including `HPOBench` and `YAHPO Gym`, under the typically used fidelity dimension, including the number of epochs, the training data sample, etc. Thanks to these benchmarks, it is also convenient to know $\zeta^{(m)}$ under different fidelities $m$ for commonly used types of fidelity dimension.
>
>     We notice that the application motivation is mentioned a bit late. In the final version, we will move the application discussion forward to the model section to better motivate the model.
>
> 2. **Discussion on regret results** We acknowledge that the results of regret minimization are present in a brief style. This is because we need to balance the presentation of two branches of contributions (best arm identification and regret minimization). We give four remarks (Remarks 4.3-4.6) in the main paper to discuss the theoretical results of regret minimization. We present the real-world implication of the two-stage arm elimination algorithm in Remark 4.3, the regret bound tightnesses in Remarks 4.4 and 4.5, and the theoretical relation to partial monitoring in Remark 4.6.
>
>     **Novelty of elimination algorithm** One novelty of the multi-fidelity elimination algorithm is its two-stage exploration-exploitation mechanism --- always conduct exploration at the highest fidelity and exploitation at the lowest fidelity. This algorithmic design is based on the special new regret definition in Eq., which reflects the real-world applications, e.g., advertisement distribution and production management, etc. (see Remarks 4.1 and 4.3 for detail).
>
>     **Tight regret bound** In Remark 4.4, we show that our problem-independent regret upper and lower bounds match up to some logarithmic factor. In Remark 4.5, we show that our problem-dependent regret upper bound matches our problem-dependent lower bound in a class of instances common in practice.
>
> 3. **Technical novelty on Theorem 3.4** Our best arm identification algorithm consists of two
>     parts: (a) LUCB framework in Algorithm 1 and (b) fidelity selection procedures in Algorithm 2. While the proof
>     approach related to LUCB framework is similar
>     to Kalyanakrishnan et al. [19], the rest proof related to fidelity selection procedures $\texttt{Explore-A}$ and -B is very different from Kandasamy et al. [19]. [19] explore the
>     fidelities from low fidelity to high fidelity one by one. While this
>     gradual elevation of the fidelity is intuitive, it fails to capture
>     that there is an optimal fidelity reflecting the best tradeoff
>     between accuracy and cost, and one should try to converge towards
>     this optimal fidelity. Instead, in our paper, we first present the
>     the lower bound result, which provides us with the above insight on the
>     optimal fidelity, and then we design two new exploration procedures
>     that try directly to converge towards optimal fidelity. Next, we
>     fix an arm $k$ and illustrate the technical challenges of analyzing
>     both procedures.
>
>     **$\texttt{Explore-A}$** utilizes a UCB-type policy to find the
>     optimal fidelity $m_k^*$ --- the most effective fidelity for
>     distinguishing this arm $k$ from the optimal arm, where the fidelity
>     corresponds to "arm" in MAB. The key step is to show that each
>     suboptimal fidelity for arm $k$ is been selected at most
>     $O(\log\log t)$ times, where $t$ is the time slot index. This
>     $\log \log t$ term, corresponding to the second term of Eq., is
>     novel and rare in the bandit's sample complexity upper bound. This
>     result leverages two crucial findings: firstly, the arm $k$ is
>     pulled a maximum of $O(\log t)$ times, as ensured by the LUCB
>     framework across all fidelities. Secondly, the
>     $\texttt{Explore-A}$ guarantees that each suboptimal fidelity is
>     chosen a logarithmic number of times *relative* to the total number
>     of arm pulling by LUCB (see
>     Lemma C.2).
>
>     **$\texttt{Explore-B}$** aims to find a good fidelity that is at
>     least half as good as the optimal fidelity. To achieve that, one key
>     technical challenge is to devise a commitment condition
>     (Line 10 in
>     Algorithm 2) and to prove that (1)
>     how much cost is sufficient to make sure the condition holds and (2)
>     when this condition holds, $\texttt{Explore-B}$ can find a good
>     fidelity with high probability (see Lemma C.3).
>
> 4. **On Experiment.** We are running more experiments, in particular comparing to the baseline LUCB on the highest fidelity, and we hope the experiments will conclude in a few days. Once it finishes we will post it as a reply. For now, please refer to Figure 1 for an empirical comparison of our proposed two exploration policies.
>
> 5. Notice that $\tilde{G}=O(M\tilde{H})$ is an upper bound for $\tilde{G}$, meaning $\tilde{G}$ is not large. That is, the second term containing $\tilde{G}$ of the cost complexity upper bound of $\texttt{Explore-A}$ is not large. Therefore, when $\tilde{G}=O(M\tilde{H})$, the cost complexity upper bound of $\texttt{Explore-A}$ is *not* far away from the lower bound. We note that the first term of $\texttt{Explore-A}$ matches the lower bound, and the second upper bound matches up to an $M$ factor.
>
>     For when do we prefer $\texttt{Explore-A}$ then $\texttt{-B}$, we kindly refer the reviewer to [the reply to review zHKa's first *Question*](https://openreview.net/forum?id=oi45JlpSOT&noteId=urNfm5ayDK).

---

> > ### Comment · Reviewer_rkrs · 2023-08-17
> > **Further clarification questions**
> >
> > I thank the authors for their response. I have a few clarification questions which will help me understand the paper:
> > - Thank you for clarifying the motivation of the paper.
> > - I still feel that the presentation of the paper is unsatisfactory where half the result, including the regret algorithm, regret theorem is in the appendix. Regarding novelty of the regret, I have a few queries.
> >      - I am concerned about the regret definition in eq (9). There are two issues. Why is the definition only discounting $\mu^1$ by $\lambda^{1}$ and not $\mu\_{I_t}$ by also $\lambda^{I_t}$? Why the $N$ is required and what happens if you simply sum over $t=1$ to $\Lambda$?
> >      - The algorithmic novelty of MF-MAB in Appendix D is still not clear to me. It seems just a kernelized version of UCB-Improved of [Auer et al  (2010)](). Pull each arm in the active set equal number of times at highest fidelity and then eliminate sub-optimal arms at that are below the gap $2^{-p + 1}$.
> >      - The factor $\epsilon$ is not described in the algorithm description in Appendix D.2 (from lines 617-622). Is it a problem dependent parameter, a lower bound to the gaps? I see that it is showing up in the Theorem D.3 as $1/\epsilon^2$. Does this mean that you need to know a lower bound to the gaps for running MF-MAB?
> >      - This queries again bring me to one of the key points I am raising. Algorithm design is one of the key contribution of this paper. You *cannot* put the algorithm in Appendix, hide away key design choices and discuss result in remarks.
> >
> > - Experiments: No further experiments are provided by the authors.
> >
> > UCB revisited: Improved regret bounds for the stochastic multi-armed bandit problem Peter Auer 2010

---

> > > ### Author Response · Authors · 2023-08-18
> > >
> > > We highly appreciate the reviewer for spending time evaluating this work.
> > >
> > > > I am concerned about the regret definition in eq (9). There are two issues.
> > > - > Why is the definition only discounting $\mu_1$ by $\lambda^{(1)}$ and not $\mu_{I_t}$ by also $\lambda^{(I_t)}$?
> > >
> > > **Answer:** We are *not* discounting reward by fidelity. No matter which fidelity is leveraged to pull the arm, the obtained observations are counted as rewards without any scaling.
> > > Please refer to Remark 4.1 for the application motivation of not discounting rewards by fidelity.
> > >
> > > Since there is no reward scaling due to different fidelity, in the second term $\mathbb{E}\left[ \sum_{t=1}^N \mu_{I_t}^{(M)} \right]$ of Eq. (9), the $\mu_{I_t}^{(M)}$ is not discounted. The first term $\frac{\Lambda}{\lambda^{(1)}}\mu_1$ of Eq. (9) signifies that the optimal strategy for maximizing cumulative rewards involves consistently pulling the optimal arm $\mu_1$ with the lowest fidelity $\lambda^{(1)}$, where the expected pulling times are $\frac{\Lambda}{\lambda^{(1)}}$, and the $\lambda^{(1)}$ is not for discounting the reward mean $\mu_1^{(M)}$.
> > >
> > > - > Why the $N$ is required and what happens if you simply sum over $t=1$ to $\Lambda$?
> > >
> > > **Answer:** $N$ represents the overall count of decision rounds (time slots) within the algorithm under consideration, and $\Lambda$ stands for the total budget allocation for said algorithm. In light of the fact that the second term in the regret definition (e.g., Eq. (9)) is the accumulation of rewards across all time slots, one needs to perform the summation across the range from $t=1$ to $t=N$.
> > >
> > > As $t$ serves as the index for time slots and possesses distinct units from the budget $\Lambda$, it is inappropriate to execute a summation over the interval $t=1$ to $t=\Lambda$.
> > >
> > >
> > > > The algorithmic novelty of MF-MAB in Appendix D is still not clear to me. It seems just a kernelized version of UCB-Improved of Auer et al (2010). Pull each arm in the active set equal number of times at highest fidelity and then eliminate sub-optimal arms at that are below the gap $2^{-p + 1}$.
> > >
> > > **Answer:** The primary innovation in the algorithm's design lies in its two-stage approach: a high-fidelity exploration phase (Lines 3-7 in Algorithm 4) followed by a low-fidelity exploitation phase (Line 8 in Algorithm 4).
> > >             As acknowledged in the paper, the design of this algorithm, particularly its exploration stage, bears resemblance to the work of Auer et al. (2010). However, our paper extends beyond this algorithmic design and centers on demonstrating that the algorithm can attain tight regret upper bounds (both problem-dependent and -independent) for the novel regret definition in Eq. (9) for the MF-MAB model.
> > >             This novel MF-MAB regret definition is suitable to model many practical application scenarios in real-world applications, e.g., advertisement distribution as described in the paper and appreciated by the other reviewers. The two-stage algorithm provides a practical strategy for such scenarios. The proposed tight regret guarantees of the two-stage algorithm support the viability of adopting such a design in practical settings.
> > >
> > > > The factor $\epsilon$ is not described in the algorithm description in Appendix D.2 (from lines 617-622). Is it a problem dependent parameter, a lower bound to the gaps? I see that it is showing up in the Theorem D.3 as $1/\epsilon^2$. Does this mean that you need to know a lower bound to the gaps for running MF-MAB?
> > >
> > > **Answer:** No, $\epsilon$ is not gap dependent and one can run the algorithm with any $\epsilon > 0$. The regret upper bound in Theorem D.3. (Eq. (21)) holds for any $\epsilon > 0$. Theorem D.4. shows that setting $\epsilon = (K\log \Lambda / \Lambda)^{1/3}$ guarantees the problem-independent (worst case) regret upper bound for any MF-MAB instance.
> > >
> > > > This queries again bring me to one of the key points I am raising. Algorithm design is one of the key contribution of this paper. You cannot put the algorithm in Appendix, hide away key design choices and discuss result in remarks.
> > >
> > > **Answer:** We thank the reviewer for suggesting a better presentation of this paper. Although we agree that the algorithm design (Algorithm 4) is a contribution to this paper, it is only one of the key contributions (among other algorithm designs and theoretical results). Especially as this is a theoretical paper, it is also very important to present the theoretical results.  We indeed made hard decisions on what to keep in the paper in order to achieve a balance.  As the camera-ready version allows an additional page, we promise to move the algorithm design and the detailed theorems in Appendix D into the main paper in the final version.

---

> > > ### Author Response · Authors · 2023-08-20
> > > **New experiment updated in general comment**
> > >
> > > This is a gentle reminder to the reviewer that we added [a new simulation in the general comments](https://openreview.net/forum?id=oi45JlpSOT&noteId=qMzrbHt8FZ). It compares our two exploration policies to one baseline and illustrates when our exploration policies (utilizing low fidelity) outperform the baseline (not utilizing low fidelity).

---

> > > > ### Comment · Reviewer_rkrs · 2023-08-21
> > > > **Response to authors**
> > > >
> > > > Thanks for your response and I have noted the simulation as well. We will discuss these results and your responses in the reviewer-area chair discussion period.

---

### Author Response · Authors · 2023-08-19
**New Experiment: Empirical Comparison to Baseline**

We highly appreciate the reviewers and area chairs for spending time evaluating this paper.

In this post, we present a new numerical simulation that compares our two exploration policies, $\texttt{Explore-A}$ and $\texttt{Explore-B}$, with a $\texttt{Baseline}$ that is always selecting the highest fidelity to explore arms. All three policies are employed under the LUCB framework (Algorithm 1). We note that, for the MF-MAB model, there is no prior algorithm (other baselines) on the best arm identification with fixed confidence setting (see Section 1.2's second paragraph for related work discussion).
This simulation aims to illustrate when our exploration policies $\texttt{Explore-A}$ and $\texttt{Explore-B}$ (utilize low-fidelity) are better than $\texttt{Baseline}$ (not utilize low-fidelity).

**Experiment Setup.** We consider a MF-MAB model consisting of $K=5$ arms and $M=5$ fidelities. The reward means are set as $\mu_{k}^{(m)} = 0.1 (k+4) + 0.01(m-5)$, where, for example, $\mu_5^{(5)} = 0.9$ is the true reward mean of optimal arm and its low-fidelity reward means are $\{0.89, 0.88, 0.87, 0.86\}$ accordingly, etc. The error upper bound in $\zeta^{(m)}$ are $\\{0.04, 0.03, 0.02, 0.01, 0\\}$ for $m\in\\{1,2,3,4,5\\}$ respectively. We set the cost of fidelity $m\in\\{1,2,3,4\\}$ as $\lambda^{(m)} = m$, and consider four different cases $I\in \\{1,2,3,4\\}$ of the cost of highest fidelity $\lambda^{(5)}=5I$. That is, from Case 1 to Case 4, the cost of the highest fidelity $\lambda^{(5)}$ increases from $5$ to $20$.

We set the confidence parameter $\delta=0.1$ and run each experiment for 100 trials. Averaged over these $100$ trials of each experiment, we present the empirical cost complexity in the table as follows,

| Cost Complexity ($\times 10^4$)           | Case 1 ($\lambda^{(5)}=5$) | Case 2 ($\lambda^{(5)}=10$) | Case 3 ($\lambda^{(5)}=15$) | Case 4 ($\lambda^{(5)}=20$) |
| -------------------- | ------ | ------ | ------ | ------ |
| $\texttt{Explore-A}$ |  $20.5$      |  $20.0$     |   $17.4$     | $17.4$       |
| $\texttt{Explore-B}$ |  $36.1$      |   $47.9$     |  $59.5$      |  $69.4$     |
| $\texttt{Baseline}$             | $21.4$      | $42.0$      |$64.6$       |$86.2$        |


**Discussion of Simulation Results.**
In all four cases, the cost complexities of $\texttt{Explore-A}$ are similar. Especially, when the cost of the highest fidelity $\lambda^{(5)}$ becomes larger, it is easier for $\texttt{Explore-A}$ to find out the optimal fidelity (e.g., the highest fidelity is obviously not the optimal one due to its high cost) and, hence, $\texttt{Explore-A}$ enjoys a slightly lower cost complexity. The cost complexities of both $\texttt{Explore-B}$ and $\texttt{Baseline}$ increase along these four cases.

When the cost of the highest fidelity is expensive, e.g., in Cases 3 and 4, both $\texttt{Explore-A}$ and $\texttt{Explore-B}$ outperform $\texttt{Baseline}$. This highlights the advantage of utilizing low fidelity to explore arms while avoiding using the expensive high fidelity.
When the cost of the highest fidelity is cheap, e.g., in Case 1, the cost complexity of $\texttt{Explore-A}$ is similar to $\texttt{Baseline}$, while $\texttt{Explore-B}$ is worse than $\texttt{Baseline}$. This suggests that when the cost of the highest fidelity is similar to that of lower fidelity, utilizing the high-fidelity to explore arms (i.e., $\texttt{Baseline}$) may be preferable.

In this comparison, $\texttt{Explore-A}$ always outperforms $\texttt{Explore-B}$. For another scenario where $\texttt{Explore-B}$ is better than $\texttt{Explore-B}$, please refer to Figure 1(b) in our paper.

Last but not least, we emphasize that this paper is a theoretical work and our focus is on devising algorithms and proving their theoretical guarantees.  Our numerical experiments so far (including Figures 1(a) and 1(b) in the paper and the additional one reported here) achieve the purpose of providing the basic support and verification of the results in the theorems.
We will include the new comparison in the above table to the final version of this paper as well.
We also plan to further validate our algorithms with more experiments on larger datasets.

---

### Decision · Program_Chairs · 2023-09-21

**Decision:**

Accept (poster)

**Comment:**

This paper presents a new look at the BAI and Regret-Minimization objectives of the MF-MAB problem. The complexity lower bound presented is interesting and so is the cost definition.

The knowledge of bounds on the rewards of the best two arms (at the highest fidelity) raised some concerns, but the authors make a case that this is at least reasonable in several applications. Further, given that this assumption allows for progress on this problem, I think this is okay.

I think the presentation of the paper can be improved. The authors have been given several suggestions already and should incorporate these. I think the presentation in the appendix also makes things a little hard to parse.
Here is a suggestion: Write the appendix in a more self-contained way. This helps readers significantly. For instance, please repeat the theorem statements again (along with some context) in the appendix and then present the proof. This allows for reading without constantly having to jump between documents. I also think it may be worth it to repeat the main part of the paper in the appendix (i.e., present the whole paper in supp materials) as this helps readers who use hyperlinks to jump between sections.